# β-actin dependent chromatin remodeling mediates compartment level changes in 3D genome architecture

Syed Raza Mahmood [1,2], Xin Xie [1,3], Nadine Hosny El Said [3], Tomas Venit[3], Kristin C. Gunsalus [1,4] & Piergiorgio Percipalle [3,5✉]

β-actin is a crucial component of several chromatin remodeling complexes that control chromatin structure and accessibility. The mammalian Brahma-associated factor (BAF) is one such complex that plays essential roles in development and differentiation by regulating the chromatin state of critical genes and opposing the repressive activity of polycomb repressive complexes (PRCs). While previous work has shown that β-actin loss can lead to extensive changes in gene expression and heterochromatin organization, it is not known if changes in β-actin levels can directly influence chromatin remodeling activities of BAF and polycomb proteins. Here we conduct a comprehensive genomic analysis of β-actin knockout mouse embryonic fibroblasts (MEFs) using ATAC-Seq, HiC-seq, RNA-Seq and ChIP-Seq of various epigenetic marks. We demonstrate that β-actin levels can induce changes in chromatin structure by affecting the complex interplay between chromatin remodelers such as BAF/ BRG1 and EZH2. Our results show that changes in β-actin levels and associated chromatin remodeling activities can not only impact local chromatin accessibility but also induce reversible changes in 3D genome architecture. Our findings reveal that β-actin-dependent chromatin remodeling plays a role in shaping the chromatin landscape and influences the regulation of genes involved in development and differentiation.

[1] Center for Genomics and Systems Biology, New York University Abu Dhabi (NYUAD), Abu Dhabi, United Arab Emirates. [2] Department of Biology, New York University, New York, NY, USA. [3] Program in Biology, Division of Science and Mathematics, New York University Abu Dhabi (NYUAD), Abu Dhabi, United Arab Emirates. [4] Department of Biology, Center for Genomics and Systems Biology New York University, New York, NY, USA. [5] Department of Molecular Biosciences, The Wenner-Gren Institute, Stockholm University, Stockholm, Sweden. ✉email: pp69@nyu.edu

Eukaryotic genomes are folded into a multi-layered hierarchical structure and this organization plays critical roles in development, differentiation, and gene regulation. 3D genome architecture refers to the organization of the genome into chromosomal territories, megabase-scale A and B compartments, and topologically associating domains (TADs)[1]. While this organization is regulated by architectural proteins and chromatin remodelers, recent work highlighting the role of cytoskeletal proteins in epigenetic regulation suggests that proteins like actin may also influence higher-order chromatin structure[2–6]. Regulatory mechanisms via which cytoskeletal proteins may potentially affect 3D genome architecture, however, are yet to be elucidated. We have previously shown that β-actin knockout MEFs exhibit widespread changes in heterochromatin organization[2,7] while retaining cytoskeletal functions and conditional migratory capacity due to compensatory upregulation of other actin isoforms[2,8]. The epigenetic and transcriptional changes observed in these cells are induced by genome-wide loss of chromatin binding of the β-actin-associated BAF complex subunit BRG1[2] and are thus linked to nuclear rather than cytoskeletal functions of actin. Whether dysregulation of the nuclear functions of β-actin also impacts chromatin accessibility and higher-order genome organization is not known. Here we investigate if nuclear β-actin levels can affect 3D genome architecture by influencing the complex relationship between chromatin remodeling proteins like BAF and PRCs.

BRG1 is an essential ATPase subunit of the mammalian BAF complex (SWI/SNF complex in yeast, and BAP complex in *Drosophila*) and requires β-actin for maximal ATPase activity[9]. The BAF complex plays a crucial role in development and differentiation by directly regulating the chromatin state of critical genes[10] and by opposing the repressive activity of PRCs[11]. It has been shown that BAF recruitment to specific genomic loci leads to the eviction of polycomb proteins, while removal of BAF reverts these loci to a polycomb-mediated inactive state[11]. The continuous opposition between BAF and PRCs, therefore, provides epigenetic plasticity for regulating the chromatin state of developmentally important genomic regions. Furthermore, PRCs are also involved in the maintenance of 3D genome structure[12]. Polycomb repressive complex 1 (PRC1), for instance, is thought to induce clustering of polycomb-bound chromatin and nucleosomes to form phase-separated 'polycomb bodies' which maintain long-range chromatin interactions[13,14]. Similarly, knockdown of EZH2, the catalytic subunit of polycomb repressive complex 2 (PRC2), leads to dispersion of such polycomb bodies[15]. It is therefore conceivable that any dysregulation of the BAF–PRC relationship induced by changes in β-actin levels could also influence higher-order chromatin structure.

While our previous work demonstrated that loss of β-actin disrupts BRG1 chromatin binding[2], it is not known if β actin-mediated changes in BRG1 occupancy alter the balance between BAF and PRC activities and possibly impact 3D chromatin organization. A potential relationship between β-actin and BAF/PRC-mediated chromatin remodeling raises the possibility that nuclear β-actin levels may play a regulatory role in shaping the chromatin landscape during development. Such a function is consistent with the well-established roles of actin in transcriptional reprogramming[16] and our previous work showing that β-actin knockout cells exhibit specific defects in gene regulation during neural reprogramming[7]. We, therefore, hypothesize that regulation of nuclear β-actin levels may potentially influence both local chromatin accessibility and higher-order genome organization. To test this hypothesis, we have performed a comprehensive genomic analysis of β-actin knockout MEFs using ATAC-Seq, ChIP-Seq, RNA-Seq, and HiC-seq.

## Results

### β-actin loss induces dosage-dependent changes in chromatin accessibility.
β-actin knockout cells exhibit genome-wide loss of BRG1 chromatin binding, changes in the level and localization of heterochromatin and dysregulation of genes involved in differentiation[2]. In order to test if these changes are directly related to the disruption of BAF functionality caused by β-actin loss, we first immunoprecipitated β-actin and BAF subunit BRG1 from nuclear and cytoplasmic fractions of β-actin wildtype (WT) and knockout (KO) MEFs (Fig. 1A). The results showed that although in the WT condition β-actin is found in both the nucleus and cytoplasm, BRG1 can only be co-precipitated with β-actin from the nuclear fraction of WT MEFs and not from β-actin KO MEFs. These observations indicate that β-actin and Brg1 are part of the same nuclear complex and that the association of BRG1 with β-actin is disrupted in the KO condition. Having confirmed the link between β-actin and BRG1, we then investigated how the loss of this interaction affects the chromatin remodeling activities of the BAF complex by performing ATAC-Seq on WT, KO, and β-actin heterozygous (HET) MEFs. β-actin KO cells exhibit close to zero β-actin expression both at the mRNA and protein level while HET cells show expression levels intermediate between WT and KO cells[8] (Fig. 1B). To obtain a global picture of chromatin accessibility changes induced by β-actin loss, we first performed a differential peak calling analysis between WT and β-actin KO cells. We identified 442 and 575 peaks, respectively, showing more than twofold increase or decrease in accessibility between WT and KO cells (Fig. 1C). Notably, whereas a majority of peaks showing altered accessibility overlapped intergenic or intronic regions, only regions with reduced accessibility showed a sizable overlap with transcription start sites (TSSs) or gene bodies (Fig. 1D).

Since TSSs and promoters are enriched for chromatin remodelers like BAF/BRG1[17] while enhancers typically occur in intergenic and intronic regions, we then investigated the functional impact of chromatin accessibility changes at these elements by focusing on annotated enhancers[18] and promoters (regions 1000 bp upstream and 100 bp downstream of annotated TSSs) showing more than twofold change in ATAC-Seq signal between WT and KO cells. Consistent with the enrichment of TSSs in peaks losing accessibility and intergenic regions in peaks gaining accessibility, most promoters showed decreased accessibility in KO cells while most enhancers tended to gain accessibility (Fig. 1E and Supplementary Fig. 1). Notably, GO term analysis of promoters with reduced accessibility revealed enrichment of several biological processes related to cell-fate commitment and regulation of neuron differentiation (Fig. 1F). More than a quarter of these repressed TSSs also overlapped known polycomb targets[19] possibly reflecting dysregulation of the BAF/PRC relationship. Given the well-established antagonistic roles of BAF and PRCs in regulating neuronal differentiation[20–22], this result suggests that promoter-specific loss of chromatin accessibility in KO cells is possibly linked to β-actin-dependent disruption of BRG1 binding.

In contrast, GO term analysis of genes linked to enhancers with increased accessibility in KO cells revealed enrichment of biological processes related to the skeletal system and vasculature development (Fig. 1F). As BRG1 knockdown has previously been linked to both gene repression and activation[17], it is likely that the activation of these regions is also a downstream effect of BRG1 loss. Furthermore, since BRG1 interacts with repressive chromatin remodeling complexes such as NuRD (Nucleosome remodeling and deacetylase)[23,24] and transcriptional repressors such as REST (repressor element 1-silencing transcription factor)[25], the loss of BRG1 binding may potentially affect the expression of

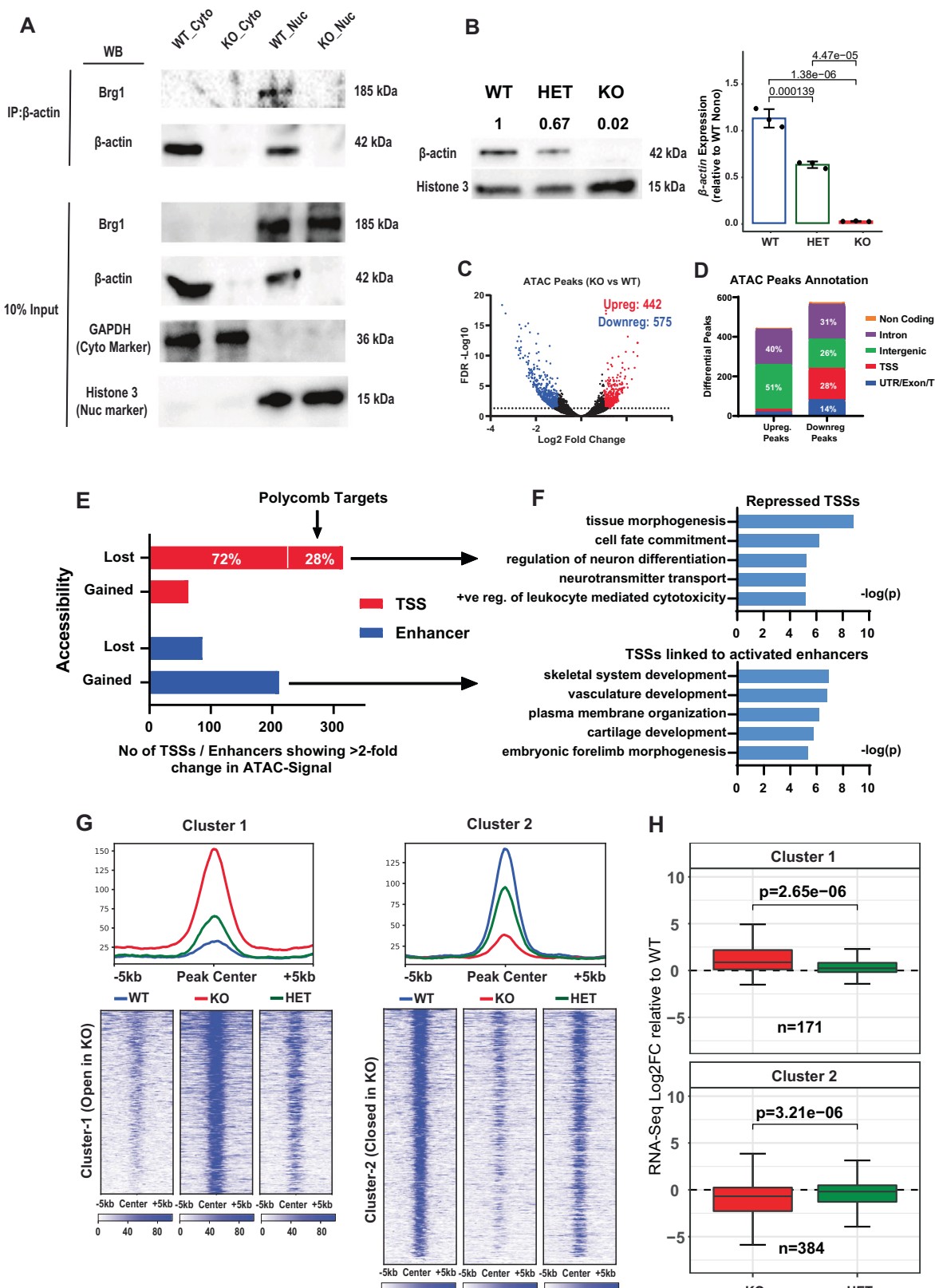

genes regulated by these complexes. Consistent with this idea, we also observe an enrichment of NuRD[26,27] and REST[28–30] regulated processed such as regulation of skeletal system development, vasculature development, and blood vessel development in the GO term analysis of genes transcriptionally upregulated in β-actin KO cells (Supplementary Fig. 2B).

To investigate if these changes in chromatin accessibility were related to β-actin levels we then clustered the differential ATAC-Seq peaks by read density into two groups showing increased (Cluster 1) or decreased (Cluster 2) chromatin accessibility upon β-actin depletion and compared their average signal between WT, KO and HET cells (Fig. 1G). Strikingly, for both clusters, β-actin

**Fig. 1 β-actin loss induces dosage-dependent changes in chromatin accessibility. A** Co-Immunoprecipitation analysis of Brg1 and β-actin interaction using anti β-actin antibody in a cytoplasmic and nuclear fraction of WT and KO cells. Histone 3 and GAPDH are used as nuclear and cytoplasmic markers, respectively. Brg1, which is only present in the nuclear fraction, can be co-precipitated with a nuclear pool of β-actin. Results based on a single experiment. WB western blot, IP immunoprecipitation, Cyto cytoplasm, Nuc nucleus. **B** Immunoblot analysis of β-actin protein levels in WT, HET, and KO cells showing relative density of bands normalized to histone H3 and qPCR quantification of β-actin expression relative to Nono housekeeping gene. Error bars show 95% confidence interval of mean, p-values based on one-way ANOVA with Tukey's multiple comparisons using $n = 3$ independent samples. **C** Volcano plot of all ATAC-Seq peaks in WT, KO, and HET cells with blue and red dots showing differential ATAC-Seq peaks whose read density increased or decreased by twofold or more (FDR < 0.05), respectively, in KO vs WT cells. Numbers in the upper right corner show number of peaks showing increased (blue) or decreased (red) accessibility in KO cells compared to WT (FDR > 0.05). p-values based on two-tailed Wald test corrected for multiple testing using Benjamini–Hochberg procedure. **D** Annotation of differential peaks gained or lost upon β-actin loss. **E** Bar graph of number of TSSs (red) and enhancers (blue) showing more than twofold change in ATAC-Seq read density in β-actin knockout versus wildtype MEFs. 28% of all TSSs losing accessibility were known polycomb targets. **F** GO term analysis of all TSSs showing >2-fold decrease in accessibility (top). GO term analysis of TSSs linked to enhancers showing >2-fold increase in accessibility (bottom). p-values based on the one-tailed hypergeometric test. **G** k-means clustering analysis of ATAC-Seq read density in differential peaks. Reads are centered on the middle of differential peaks ±5 kb and sorted by WT ATAC signal. Scale bar shows normalized RPKM. **H** Average $\log_2$ fold change with respect to wildtype MEFs of protein-coding genes overlapping repressed and activated clusters. Boxes represent first and third quartiles with line in the box showing median and whiskers showing data within 1.5× interquartile range. p-values based on two-tailed, two-sample Wilcoxon-rank sum test. $n$ = Number of genes in each group. Source data are provided as a Source Data file.

heterozygous cells exhibited chromatin accessibility intermediate between WT and KO cells (Fig. 1G). This relationship was also observed at the level of transcription as genes overlapping open and closed clusters showed a smaller magnitude of up or downregulation in HET cells as compared to KO cells (Fig. 1H). Such dosage dependence of chromatin accessibility on β-actin levels was not observed for a randomly selected set of ATAC-Seq peaks (Supplementary Fig. 3A). We also did not observe a significant difference in the pattern of nucleosome occupancy at gene promoters in WT and KO cells indicating that the loss of β-actin has little impact on genome-wide nucleosome positioning (Supplementary Fig. 3B). Our data, therefore, reveal a direct relationship between β-actin levels and chromatin accessibility of specific genomic regions.

**Chromatin accessibility changes in β-actin KO cells correlate with dysregulation of BRG1- and EZH2-mediated chromatin remodeling**. Since our data suggested that differences in chromatin accessibility between WT and KO cells were potentially linked to dysregulation of BAF/PRC-mediated epigenetic changes, we then analyzed the epigenetic landscape of these cells. We performed ChIP-Seq analysis of PRC2 methyltransferase subunit EZH2 and integrated these results with previously published ChIP-Seq data for BRG1, H3K9me3, and H3K27me3[2]. BRG1 and EZH2 are expressed at similar levels in WT and KO cells (Supplementary Fig. 4A, B) and we have previously shown that these cells also do not show significant differences in the expression of various histone-methyltransferases and polycomb group proteins[2]. In order to obtain a broad overview of the relationship between the different epigenetic marks we first performed an analysis of the pairwise correlation between BRG1, EZH2, H3K9me3, and H3K27me3 in WT and KO cells. Our results revealed important differences in the epigenetic landscape of WT and KO cells with BRG1, EZH2, and H3K27me3 being much more highly correlated with each other in KO cells as compared to WT cells (Supplementary Fig. 4D). As this result potentially signified dysregulation of BAF/PRC activity, we further explored this idea by generating average ChIP-Seq profiles of different epigenetic marks across all TSSs (Fig. 2A). As previously reported[2], KO cells showed a genome-wide loss of BRG1 chromatin binding at TSSs (Fig. 2A-i). In agreement with the antagonistic relationship between BRG1 and EZH2 chromatin binding, loss of BRG1 was accompanied by a global increase in the levels of EZH2 (Fig. 2A-ii) and its associated epigenetic mark H3K27me3 at all TSSs (Fig. 2A-iv). Notably, unlike the genome-

wide changes observed in BRG1, EZH2, and H3K27me3 levels, changes in the constitutive heterochromatin mark H3K9me3 were limited to specific subsets of TSSs as indicated by a slight bump in ChIP-Seq signal centered around TSSs in KO cells (Fig. 2A-iii).

We then sought to understand the mechanistic relationship between the observed epigenetic changes and chromatin accessibility by analyzing ChIP-Seq signal for each epigenetic mark across the previously described ATAC-Seq clusters (Fig. 2B). We observed a strong relationship between H3K9me3 and chromatin accessibility as the open cluster (Cluster-1) also showed a significant loss of H3K9me3, while the closed cluster (Cluster-2) showed a significant gain. Our data also revealed a relationship between chromatin accessibility and the extent of BRG1 loss. The more accessible cluster (Cluster-1) appeared relatively resistant to β-actin-mediated BRG1 loss while the less accessible cluster showed a pronounced loss of BRG1 binding (Cluster-2). In striking contrast to BRG1 and H3K9me3 patterns, EZH2 and its associated modification H3K27me3 accumulated in both clusters.

To understand the functional significance of widespread EZH2 accumulation in both ATAC-Seq clusters as well as at TSSs, we performed a differential analysis of EZH2 peaks between WT and KO cells. Our results revealed a genome-wide increase in EZH2 levels with a negligible number of regions showing EZH2 loss (Fig. 2C). While this increase was largely confined to intergenic and intronic regions, a considerable number of TSSs also gained EZH2 peaks (Fig. 2D). Similar to our observations for sites showing increased accessibility in KO cells, GO term analysis of regions gaining EZH2 peaks revealed an enrichment of biological processes linked to the skeletal system and vasculature development (Fig. 2F and Fig. 1E). As previously discussed, genes associated with these processes are also transcriptionally upregulated in KO cells (Supplementary Fig. 2B). Importantly, these biological processes are known to be activated by EZH2 via a polycomb- and methylation-independent role as a transcriptional activator[31,32] (Supplementary Fig. 2D). Our data, therefore, suggest that in addition to its role as a global repressor, EZH2 is also present at transcriptionally active regions in KO cells and is potentially involved in gene activation.

As our results hinted at a dual role for EZH2 in both gene activation and gene repression, we then wondered why EZH2 accumulation correlated with increased chromatin accessibility in one ATAC-Seq cluster and decreased chromatin accessibility in the other. To understand the relationship between EZH2

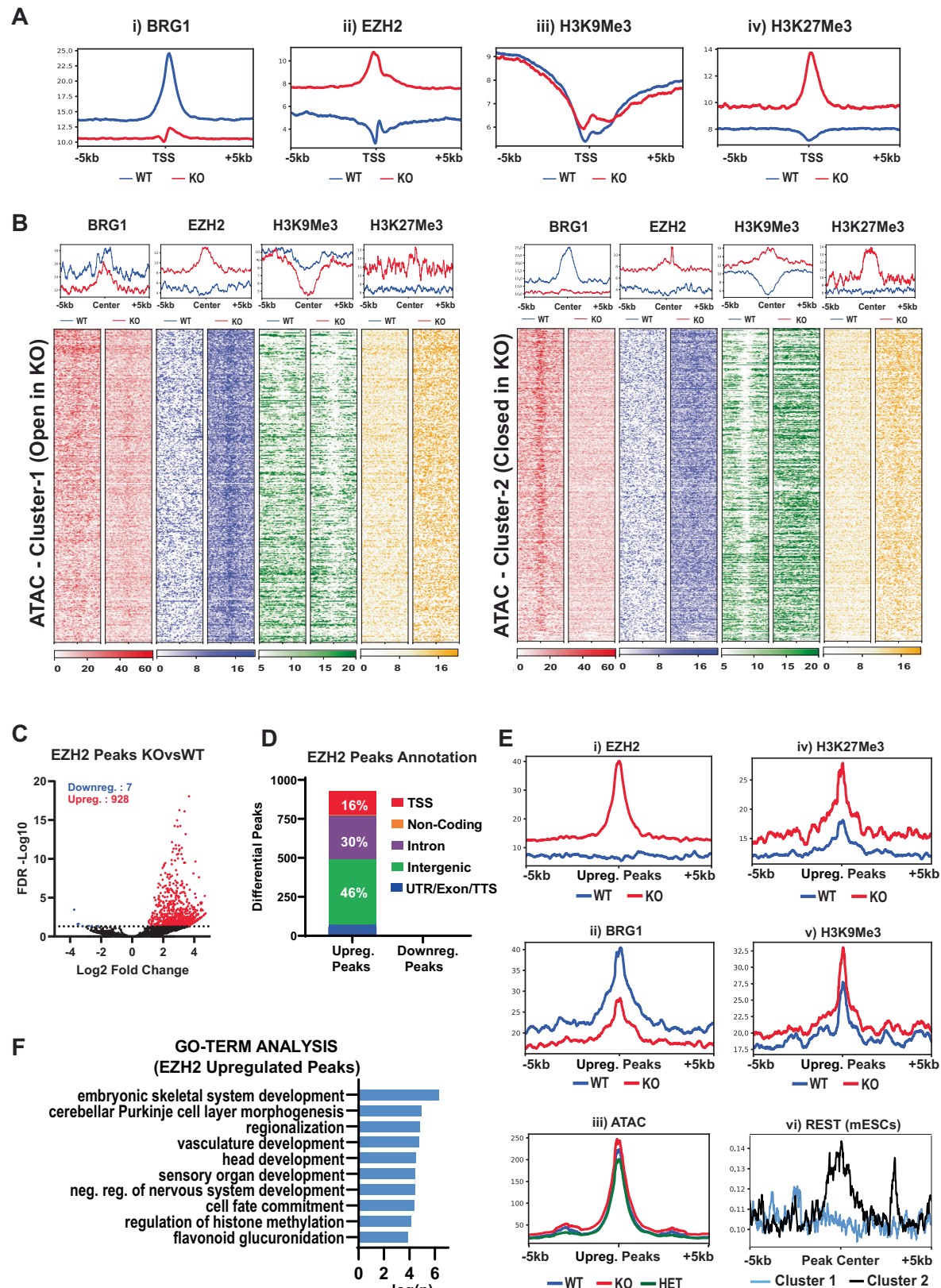

accumulation, chromatin accessibility, and the epigenetic landscape, we focused on EZH2 peaks showing a significant increase in EZH2 levels and plotted ATAC-Seq and ChIP-Seq signal for each epigenetic mark in these regions (Fig. 2E). As expected, these regions also showed significant BRG1 loss (Fig. 2E-ii) and H3K27me3 accumulation (Fig. 2E-iv). However, EZH2 gaining

regions did not show an overall change in ATAC-Seq signal (Fig. 2E-iii), suggesting that the presence of EZH2 alone is not sufficient for inducing loss of chromatin accessibility.

This result is consistent with a recent study showing that while loss of BAF activity leads to increased EZH2 recruitment, this only induces repression at a subset of genes where EZH2 colocalizes

**Fig. 2 β-actin loss induces widespread changes in the epigenetic landscape. A** Read density profiles showing average signal intensities (RPKM) ± 5 kb of all TSSs in the genome for (i) BRG1, (ii) EZH2, (iii) H3K9me3, and (iv) H3K27me3. **B** Density plots showing average signal intensities (top) and heatmaps displaying scaled-read densities (bottom) for BRG1 (red), EZH2 (blue), H3K9me3 (green), and H3K27me3 (yellow) at regions surrounding ±5 kb of opened (left) and closed (right) ATAC-Seq clusters. All plots are sorted by WT ATAC signal. Scale bar shows normalized RPKM. **C** Volcano plot of all EZH2 peaks in WT and KO cells with red and blue dots showing differential peaks whose read density increased or decreased by twofold or more (FDR < 0.05), respectively, in KO vs WT cells. Numbers in the upper left corner show number of peaks with twofold increase (red) or decrease (blue) in read density (FDR > 0.05) *p*-values based on two-tailed Wald test corrected for multiple testing using Benjamini–Hochberg procedure. **D** Annotation of differential EZH2 peaks gained or lost upon β-actin loss. **E** Read density profiles showing average signal intensities (RPKM) ± 5 kb around the center of EZH2 peaks gained in KO cells for (i) EZH2, (ii) BRG1, (iii) ATAC, (iv) H3K27me3, (v) H3K9me3, and for (vi) published REST ChIP-Seq data in mESCs showing log2FC over control for peaks in cluster 1 (blue) and cluster 2 (black). **F** GO term analysis of genes overlapping EZH2 peaks gained in KO cells. *p*-values based on a one-tailed hypergeometric test.

with other repressors such as REST[33]. Regions lacking such co-repressors on the other hand maintain sufficient residual levels of BAF binding to oppose EZH2-mediated repression[33]. Consistent with this model, while EZH2 accumulated uniformly across all differentially accessible regions in KO cells (Fig. 2B), regions showing increased accessibility (Cluster 1) not only retained a residual level of BRG1 binding but also showed transcriptional upregulation despite a pronounced increase in EZH2 levels (Figs. 1H and 2B). An analysis of published ChIP-Seq data for REST[34] further confirmed that Cluster 2, which exhibited reduced accessibility, was also highly enriched for REST binding in mESCs in comparison with Cluster 1 (Fig. 2E-vi). Since REST and its co-repressors can mediate long-term gene silencing by recruiting H3K9me3 histone-methyltransferases[35–37], this result also potentially explains the striking increase in H3K9me3 in Cluster 2 (Fig. 2B). Our findings, therefore, support a model where β-actin levels can influence the complex interaction between chromatin remodelers such as BRG1, EZH2, and co-repressors like REST in a dosage-dependent manner, leading to both activation and repression of developmentally important genes.

**β-actin KO cells exhibit reversible changes in 3D genome architecture at the level of compartments**. Since our ATAC-Seq and ChIP-Seq results revealed significant β-actin-dependent changes in BAF/PRC-mediated chromatin remodeling, we next investigated whether these changes could also induce alterations in higher-order chromatin organization. Accumulating evidence based on both microscopy and proximity ligation assays such as Hi-C has revealed that the eukaryotic genome is organized into a multi-layered hierarchical structure divided into chromosomal territories and megabase-scale A and B compartments[38]. Compartments represent genomic regions that exhibit high interaction frequencies with each other and low interaction frequencies with regions in other compartments. Such compartments correlate with euchromatin and heterochromatin and are either accessible, gene-rich, and transcriptionally active (A-Compartment) or inaccessible gene-poor and transcriptionally repressed (B-Compartment)[38].

To test if loss of β-actin led to changes in compartment-level chromatin organization, we conducted Hi-C sequencing on β-actin WT, KO, and HET cells. Furthermore, to test if any observed changes in genome organization in KO cells could be rescued by the reintroduction of β-actin, we also performed Hi-C sequencing on previously generated[2] KO cells expressing NLS-tagged mouse or human β-actin and control cells expressing GFP only in the KO background (hereafter referred to as KO-Mm, KO-Hs, and KO-GFP, respectively). To obtain an overall picture of changes in genome-wide interaction frequencies induced by β-actin loss, we first constructed Hi-C interaction matrices at 500 kb resolution and clustered them based on Spearman rank correlation. Our results (Fig. 3A and Supplementary Fig. 5A) showed that cells expressing NLS-tagged mouse or human actin in the KO

background ((KO-Mm and KO-Hs), respectively) clustered together while KO cells and GFP expressing negative control cells (KO-GFP) formed a separate cluster. Similarly, WT and HET cells clustered separately from all KO samples. These results showed that variation in β-actin levels produced significant differences in genomic interactions and that the similarity in contact maps of the rescue samples was directly related to β-actin expression.

We then binned the genome into 500 kb non-overlapping intervals and used principal components analysis (PCA) to classify the resulting bins as belonging to either A or B compartment (Fig. 3B). As compartment identity is reflected in the sign of the first principal component (PC1)[38], regions that switch compartments upon β-actin loss show the opposite PC1 sign in WT and KO cells. Our results revealed that 94% of the genome remained in the same compartment (40% A and 54% B), while 6% of all genomic bins switched from A to B or from B to A upon β-actin loss. These switching bins were highly consistent between biological replicates (Supplementary Fig. 5B) and we refer to these henceforth as A → B, and B → A, respectively. As expected, compartment A contained more than twice the number of genes and exhibited a much higher gene density than compartment B (Fig. 3C). Notably, genomic regions that switched compartments showed a gene density intermediate between constitutive compartments. Switching compartments also showed expected changes in the transcriptional (Fig. 4A) and epigenetic (Fig. 4B) landscape with A → B bins accumulating histone methylation and showing decreased chromatin accessibility and gene expression. These changes were also observed at the promoters of several developmentally important regulators of cell differentiation located within these regions including *Sox21*, *Bmp-3*, and *Bmp-6* (Supplementary Fig. 6A). qPCR validation of the expression of selected genes also confirmed down and upregulation of A to B and B to A switching genes, respectively (Supplementary Fig. 6C). Similarly, contact maps of switching genes such as *Bmp3*, *Fgf5*, and *Sv2c* (Supplementary Fig. 7) showed significant changes in interactions between WT and KO cells. Finally, GO term analysis of genes located in the switching compartments (Supplementary Fig. 6B) revealed enrichment of several biological processes which we have previously shown to be dysregulated in KO cells, such as mitochondrial function and response to interferon-α[39,40].

To more closely study the impact of β-actin levels on compartment-level organization, we then asked whether bins that switched compartments between WT and KO cells also showed the same switch in heterozygous cells and cells rescued with a mouse or human NLS-β-actin (Fig. 3D, E). Of the bins that switched from A → B or B → A in KO cells (Fig. 3D-i), only 42% and 50%, respectively, showed the same switch in HET cells (Fig. 3D-ii) suggesting that β-actin levels were important for mediating compartment switching. Similarly, reintroduction of NLS-tagged mouse β-actin in the KO background reversed the

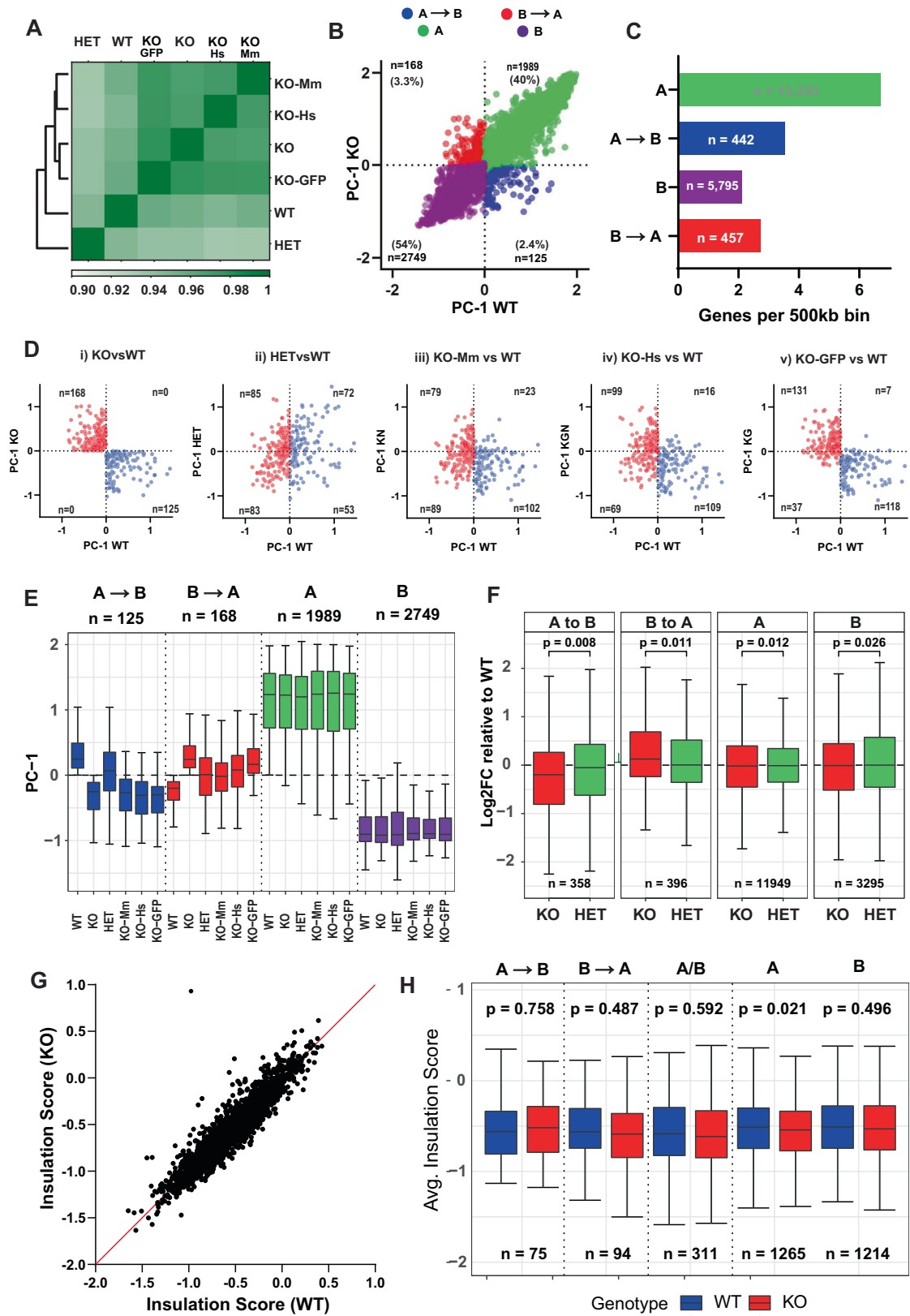

compartment switch for 18% of A → B bins and 52% of B → A bins (Fig. 3D-iii). Reintroduction of human β-actin produced similar levels of compartment rescue (Fig. 3D-iv), while negative controls expressing GFP showed little or no change (Fig. 3D-v). Unexpectedly, compartment rescue by mouse (Fig. 3D-iii) or

human (Fig. 3D-iv) NLS- β-actin was distinctly stronger for bins that were activated upon β-actin depletion (lower left quadrant) than bins that were repressed upon β-actin depletion (upper right quadrant). While the reason for this is not clear, based on our ATAC-Seq and ChIP-Seq data, we speculate that the repressed

**Fig. 3 β-actin loss induces reversible changes in 3D genome architecture. A** Clustering of genome-wide Hi-C contact matrices at 500 kb resolution. Colors represent spearman correlation - higher values correspond to higher similarity. **B** Scatter plot showing PC1 values of 500 kb genomic bins in WT (x axis) and KO (y axis) cells based on PCA analysis of Hi-C contact maps. Red and blue dots represent bins switching from B → A and A → B, respectively, green and purple dots represent non-switching A and B bins. Numbers in top and bottom left of all quadrants indicate a total number of bins in each compartment and their percentage as compared to all genomic bins. **C** Gene density bar graph showing the number of genes per 500 kb bin in each compartment. Red and blue bars represent regions switching from B → A and A → B, respectively, green and purple bars represent non-switching A and B regions. Numbers inside the bar show total number of protein-coding genes overlapping each compartment. **D** Scatter plots of genomic bins that switched from A → B (blue) or B → A (red) in KO vs WT MEFs. For each plot the x axis shows WT-PC1 value while the y axis shows PC1 value for (i) KO (β-actin knockout MEFs), (ii) HET (β-actin heterozygous MEFs), (iii) KO-Mm (β-actin knockout MEFs expressing NLS-tagged mouse β-actin), (iv) KO-Hs (β-actin knockout MEFs expressing NLS-tagged human β-actin), and (v) KO-GFP (β-actin knockout MEFs expressing GFP). n = number of bins in each quadrant. **E** Boxplots of PC1 value of 500 kb bins in different compartments for each cell type. Red and blue boxplots represent regions switching from B → A and A → B, respectively, green and purple bars represent non-switching A and B regions. Boxes represent first and third quartiles with line in the box showing median and whiskers showing data within 1.5× interquartile range. n = Number of 500 kb bins in each group. **F** Average $\log_2$ fold change in expression for all protein-coding genes overlapping each compartment with respect to wildtype MEFs. Red boxplots show wildtype versus β-actin knockout expression change, green boxplots show wildtype versus β-actin heterozygous expression change. Boxes represent first and third quartiles with line in the box showing median and whiskers showing data within 1.5× interquartile range. p-values based on two-tailed, two-sample Wilcoxon-rank sum test. n = Number of genes in each group. **G** Insulation scores of TAD boundaries identified at 50 kb resolution in wildtype cells on the x axis versus insulation score for the same boundaries in β-actin knockout cells on the y axis. **H** Boxplots showing insulation scores of TAD boundaries overlapping switching compartments (A → B or B → A), stable compartment (A or B), and multiple compartment (A/B). Boxes represent first and third quartiles with line in the box showing median and whiskers showing data within 1.5× interquartile range. p-values based on two-tailed, two-sample Wilcoxon-rank sum test. n = Number of TAD boundaries in each group.

bins may be more resistant to rescue due to heterochromatin formation induced by the accumulation of repressive histone methylation marks and polycomb proteins. Consistent with this idea, we find that A → B bins exhibit significant accumulation of the constitutive heterochromatin mark H3K9me3 in KO cells while B → A bins do not (Fig. 5B-ii).

Since compartmentalization of the genome is known to be correlated with gene expression, we then investigated how compartment switching affected the transcriptional landscape. Using previously published RNA-Seq data[2], we analyzed the average KO vs WT expression change for all protein-coding genes located in different compartments. As expected, average expression decreased in A → B regions and increased in B → A regions while stable compartments showed minimal changes in average expression (Fig. 3F). HET cells showed intermediate gene expression changes in switching compartments demonstrating β-actin dosage dependence at the level of transcription. Our results, therefore, confirm that compartment-level changes in chromatin organization and concomitant effects on transcription are closely linked to nuclear β-actin levels.

Genomic compartments are composed of sub-megabase scale topologically associating domains (TADs), which form micro-environments that are enriched for local chromatin interactions[41]. Previous work has shown that loss of BRG1 activity can potentially influence TAD-level organization by affecting CTCF localization and chromatin accessibility at TAD boundaries[42]. We, therefore, investigated if loss of β-actin could result in changes in TAD organization either directly or via a BRG1-mediated mechanism. TAD analysis at 50 kb resolution revealed that a majority of TAD boundaries showed little or no difference in insulation scores between WT and KO cells (Fig. 3G). Similarly, we observed no significant difference in compartment-specific TAD boundary insulation scores (Fig. 3H) or in chromatin accessibility at TAD boundaries and CTCF binding sites (Supplementary Fig. 8C, D). These results suggest that TAD organization is largely conserved upon β-actin depletion and the accompanying loss of BRG1 chromatin binding. While our results are consistent with recent work showing that loss of BRG1 has no impact on genome-wide insulation scores[34], we cannot completely discount the possibility that a smaller subset of TAD boundaries may be affected by the loss of β-actin or BRG1.

**Sequence composition and epigenetic landscape influence susceptibility to compartment-level reorganization.** As our data showed that β-actin depletion induces changes in higher-order chromatin organization at the level of A and B compartments (Fig. 3B), we wondered if the sequence composition and epigenetic landscape of switching and non-switching compartments could shed light on their susceptibility to reorganization. Since BRG1-interacting chromatin remodelers and transcriptional repressors such as EZH2, REST and NuRD are known to target regions rich in GC and CpG[24,30,43], we first analyzed the GC and CpG content of all compartments. Interestingly, regions switching from an active to repressive state (A → B) exhibited a higher GC and CpG content than non-switching regions or regions switching from repressed to active states (B → A) (Supplementary Fig. 9A). Similarly, regions showing increased EZH2 accumulation also had higher GC and CpG content than regions not showing a significant change in EZH2 levels (Supplementary Fig. 9B).

We next asked if in addition to the sequence composition, the different compartments also possessed distinct epigenetic landscapes. To explore the epigenetic differences between different compartment types, we first focused on WT cells and investigated how the pairwise Spearman correlation of all epigenetic marks changed between constitutive A and B compartments. We found that while most epigenetic marks showed relatively similar pairwise correlations in both compartments, the correlation between H3K9me3 and all other epigenetic marks was much stronger in the B (Fig. 5A-ii, bottom right triangle) versus the A compartment (Fig. 5A-i, bottom right triangle). Since H3K9me3 is a marker of constitutive heterochromatin, this result potentially reflected greater enrichment of H3K9me3 throughout the repressive B compartment (Supplementary Fig. 9C-ii). H3K9me3 also showed a clear negative correlation with chromatin accessibility (ATAC-Seq) in the A compartment (Fig. 5A-i, bottom right triangle), consistent with its well-established role of repressing lineage-inappropriate genes and acting as a barrier to cell-fate change[44]. Notably, we also observed that the correlation between chromatin accessibility (ATAC-Seq) and H3K9me3 in switching compartments (Fig. 5A-iii, iv, bottom right triangle) was intermediate between that observed in constitutive A and B compartments (Fig. 5A-i, ii, bottom right triangle). Similarly, switching regions in WT cells also showed

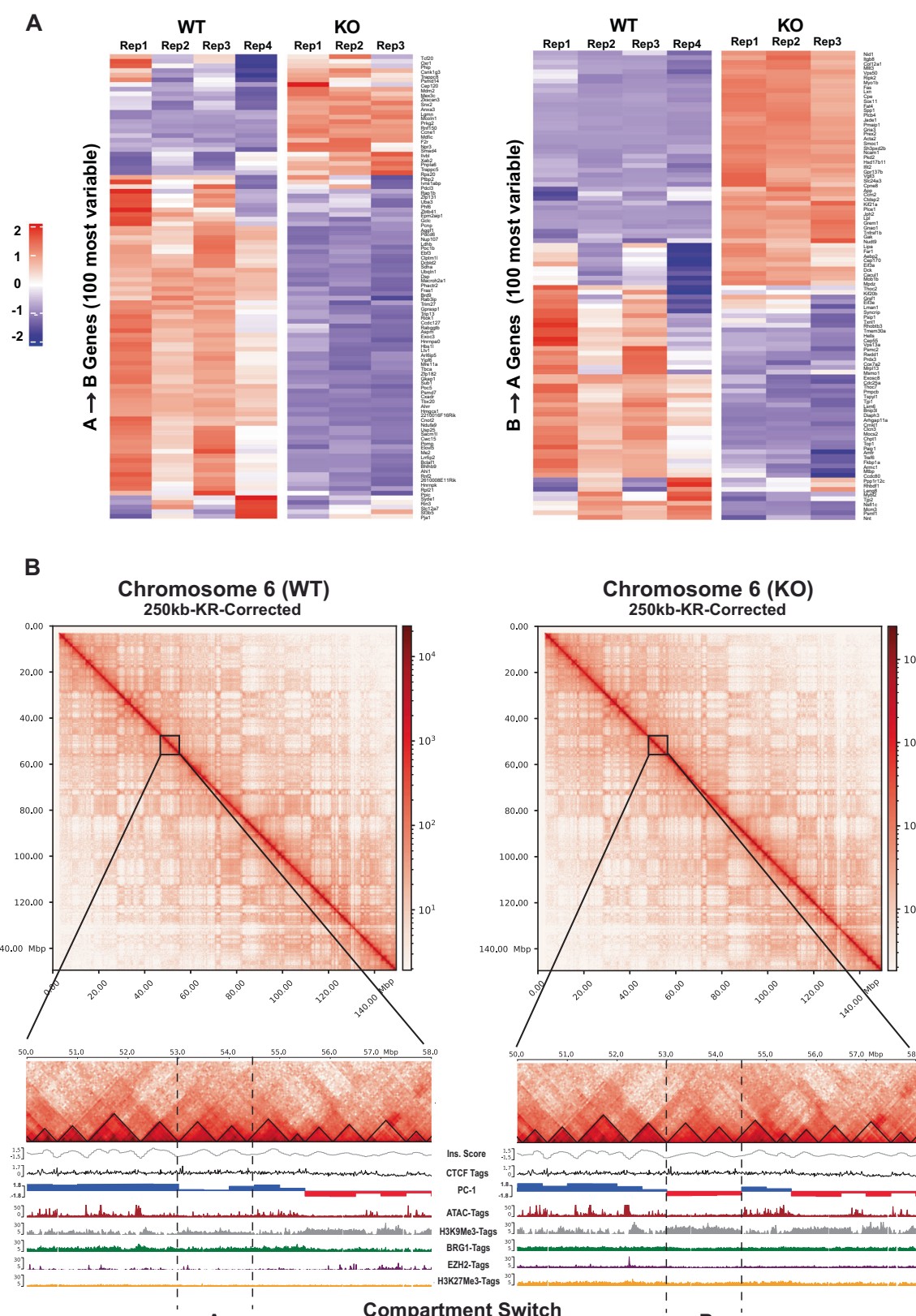

intermediate levels of chromatin accessibility (Supplementary Fig. 9A-i), H3K9me3 levels (Supplementary Fig. 9A-ii), and BRG1 occupancy (Supplementary Fig. 9A-iii) relative to stable regions. We, therefore, speculate that these regions represent an intermediate epigenetic state susceptible to reorganization. Finally, we also detected a noticeable difference between A → B

and B → A regions in the correlation of H3K9me3 and H3K27me3 with each other and with other epigenetic marks. H3K9me3 and H3K27me3 were much more highly correlated in A → B regions (Fig. 5A-iii, bottom right triangle) than in B → A regions (Fig. 5A-iii, bottom right triangle), potentially explaining why A → B regions are prone to repression. Our results,

**Fig. 4 Heatmaps showing gene expression changes and Hi-C contact maps showing epigenetic changes associated with compartment switching. A** Top 100 protein-coding genes showing the highest variance in normalized read counts (CPM) between wildtype and β-actin knockout MEFs belonging to A → B (left) and B → A (right) switching bins. Heatmap based on z-score normalized CPM. **B** Hi-C contact matrices at 250 kb resolution for chromosome 6 (top) and zoomed in view of genomic bins (bottom) switching from compartment A in WT (right) to B in KO cells (left). TAD insulation scores calculated at 50 kb resolution (gray), ENCODE CTCF density (black), PC1 values calculated at 500 kb resolution (red: negative and blue: positive), RPKM normalized ATAC-tags (brown), RPKM normalized H3K9me3-tags (gray), RPKM normalized BRG1-tags (green), RPKM normalized EZH2-tags (purple), and RPKM normalized H3K27me3 (yellow).

therefore, support the idea that both epigenetic state and sequence composition influence which genomic regions are most sensitive to chromatin reorganization induced by loss of β-actin.

**β-actin-dependent compartment switching correlates with changes in H3K9me3.** Since our results showed that different compartments possessed distinct epigenetic landscapes in WT cells, we next investigated how the loss of β-actin impacted these epigenetic states in KO cells. To identify the epigenetic changes induced by β-actin loss in each compartment, we first compared the pairwise correlation of different epigenetic marks between WT and KO cells. Our results revealed a striking increase in the correlation of BRG1, EZH2, and H3K27me3 with each other in all compartments of KO cells compared to WT cells (Fig. 5A, top left triangles). This observation highlighted the dysregulation of the BAF/PRC relationship induced by β-actin loss and potentially signified increased accumulation of EZH2 and H3K27me3 in chromatin regions normally occupied by BRG1. We, therefore, calculated the compartment-wise KO/WT log2 fold-change for each epigenetic mark in 100 kb genomic bins and our results indeed confirmed that both EZH2 (Fig. 5B-i) and H3K27me3 (Fig. 5B-ii) showed increased accumulation in the gene-rich A compartment, which is normally enriched for BRG1 (Supplementary Fig. 9C iii).

To explore the relationship between compartment switching and gain or loss of specific epigenetic marks, we then plotted the KO/WT log2 fold-change for each epigenetic mark in 100 kb genomic bins against change in compartment state as reflected by (KO-WT) PC1 value of these bins. We also repeated this analysis using 50 and 250 kb bin size (Supplementary Fig. 10) to ensure that our results were not affected by the resolution of the analysis. Surprisingly, our results revealed relatively little correspondence between BRG1 occupancy changes and compartment status (Fig. 5B-iii) in contrast to the strong relationship previously observed between BRG1 loss and loss of chromatin accessibility in KO cells (Fig. 2B, Cluster 2). Similar results were obtained for EZH2 (Fig. 5B-i) and its associated epigenetic mark H3K27me3 (Fig. 5B-ii), which showed similar levels of change across most bins irrespective of switching behavior. These results are consistent with the idea that while β-actin-mediated BRG1 loss may act as the initial trigger for chromatin reorganization, other factors such as sequence composition and epigenetic landscape also strongly influence the susceptibility of specific genomic regions to compartment switching.

The strongest correlation between compartment switching and epigenetic change was observed for H3K9me3 (Fig. 5B-iv), which seems to be the primary epigenetic mark driving both chromatin accessibility and compartment-level changes in KO cells. This result is in line with the observation that H3K9me3 is also the main epigenetic mark that differentiates A and B compartments in WT cells (Fig. 5A) and with accumulating evidence supporting the role of H3K9me3 and HP1α-mediated phase-separation in compartment-level genome organization[45–47]. Consistent with the idea of H3K9me3 mediated chromatin compaction induced by β-actin loss, KO cells also exhibited a slight reduction in the frequency of long-range interactions (Supplementary Fig. 11) and

a downregulation of DNA repeats (Supplementary Fig. 12) such as long interspersed nuclear elements (LINEs) which are known to be repressed by H3K9me3[48]. Based on our results, we, therefore, propose a model (Fig. 5C) in which changes in β-actin levels can trigger compartment-level chromatin reorganization by influencing the complex interplay between BRG1 and EZH2 chromatin remodeling activities. Such changes can lead to the gain or loss of H3K9me3 in specific genomic regions, and in combination with other factors such as initial epigenetic state, sequence composition, and presence of co-repressors like REST or HP1α, induce changes in both local and higher-order genome organization.

## Discussion

Several recent studies have raised the intriguing possibility that cytoskeletal proteins like actin could play a role in regulating higher-order genome organization[2–6]. The precise mechanisms via which cytoskeletal proteins may impact chromatin structure, however, have remained unclear. Our previous work had demonstrated that loss of β-actin induced widespread changes in heterochromatin organization by disrupting the chromatin remodeling activity of the BAF complex and hinted at the potential dysregulation of the BAF–PRC relationship[2]. It also showed that β-actin KO cells exhibit increased H3K9me3 accumulation at TSSs and localization of heterochromatin organizing protein HP1α towards the nuclear interior[2]. Since both polycomb group proteins[13,14,49–51] and HP1α[45–47] have been implicated in initiating phase-separation, a process thought to underlie the self-organization of chromatin into A and B compartments[12], we surmised that dysregulation of chromatin remodeling by these proteins in β-actin KO cells could constitute a potential pathway via which nuclear actin levels may influence 3D genome structure.

Using a comprehensive genomic analysis of β-actin KO cells, we have now demonstrated a clear link between nuclear β-actin, chromatin accessibility, and compartment-level changes in genome organization. Our results show that loss of nuclear β-actin can induce compartment-level changes in genome organization by influencing the BAF–PRC relationship and triggering the accumulation of EZH2 and H3K9Me3 in specific genomic regions. Since H3K9me3 is characteristic of HP1α positive heterochromatin and shows a stronger correlation with compartment switching than the other epigenetic marks, it is plausible that HP1α mediated phase-separation could underlie compartment-level changes induced by β-actin loss. A more detailed investigation of the physical state of chromatin regions that switch compartments in β-actin KO cells along with imaging analysis using techniques such as DNA-FISH is, therefore, an intriguing area of future research. Interestingly, while HP1α and H3K9me3 are involved in the organization of heterochromatic regions, the possibility that nuclear actin levels may also regulate euchromatic compartments cannot be overlooked. The formation of active compartments is thought to be based on phase-separation induced by multivalent interactions between RNA polymerase II, transcription factors, and chromatin regulators[12,52]. Recent work has demonstrated that such RNA

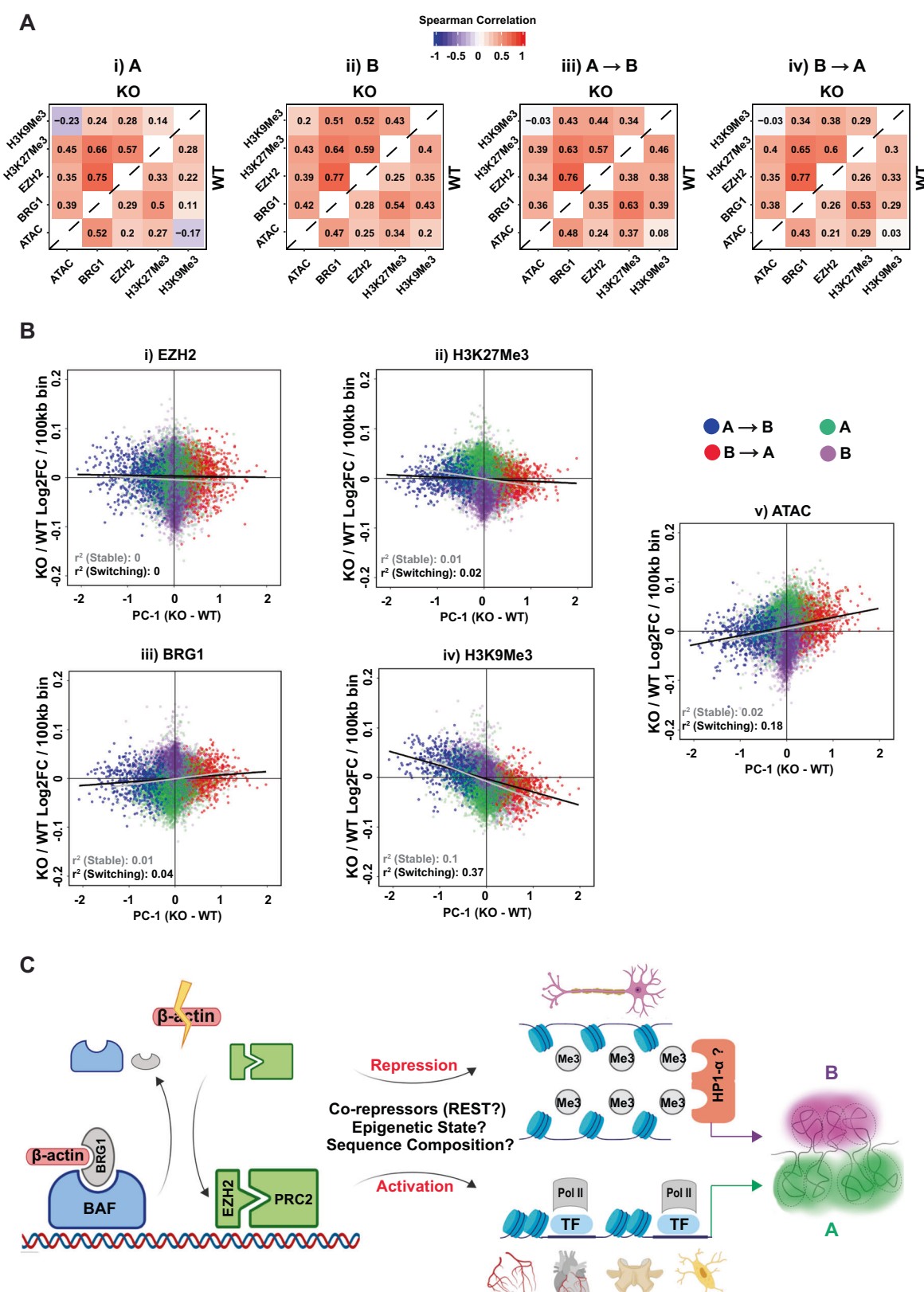

polymerase II clustering is also influenced by nuclear actin levels[53]. A possible role for nuclear actin in regulating the formation of both A and B compartments is, therefore, an exciting future research direction.

In this study, we have shown that changes in β-actin levels can have a dramatic impact on the interplay between developmentally

important chromatin remodelers. Our results demonstrate that loss of β-actin-mediated BRG1 chromatin binding induces both extensive epigenetic changes such as genome-wide increase in EZH2 and H3K27me3, and specific chromatin accessibility changes driven by H3K9me3. We show that changes in the levels of a cytoskeletal protein like β-actin can influence 3D genome

**Fig. 5 Compartment switching correlates with specific epigenetic changes. A** Pairwise spearman correlation heatmap of all epigenetic marks in compartment (i) A, (ii) B, (iii) A → B, and (iv) B → A. Top left and bottom right triangles show pairwise spearman correlations for KO and WT cells, respectively (matrices have not been clustered by similarity to facilitate comparison between compartments). **B** Scatter plots showing the relationship between PC1 value and (i) EZH2, (ii) H3K27me3, (iii) BRG1, (iv) H3K9me3, and (v) ATAC-Seq signal for each compartment. y axis shows KO over WT Log2FC in rlog/VST normalized counts for each 100 kb bin while x axis shows the difference between KO and WT-PC1 value for the same bin. R-Squared and line of best fit for switching bins only (A → B and B → A) shown in black, R-squared and line of best fit for stable bins only (A and B) shown in gray. **C** Proposed model showing the impact of β-actin loss on genome organization. Loss of β-actin disrupts BRG1 chromatin binding and induces EZH2 accumulation. Depending on the epigenetic context, sequence composition, and presence of co-repressors this leads to gene activation or repression and compartment-level changes possibly mediated by HP1α (created with BioRender.com).

architecture and also demonstrate significant dosage dependence between β-actin levels and changes in chromatin organization. Given the well-established role of SWI/SNF and PRC dysregulation in cancer[54] and neurological disorders[21], our findings also present β-actin as a regulatory factor that may potentially influence these complexes in development and disease. This study therefore provides a foundation for future experiments investigating additional functions of cytoskeletal proteins, their potential role in the organization of genomic compartments, and the regulatory mechanisms affected by their disruption.

## Methods

**Cell culture.** WT, KO, and HET mouse embryonic fibroblasts (MEFs) (from the lab of Dr. Christophe Ampe, University of Gent, Belgium) were maintained and cultured with Dulbecco's modified Eagle medium (DMEM) with high glucose, 10% fetal bovine serum (FBS) and 100 U/mL penicillin and 100 μg/mL streptomycin, in a humidified incubator with 5% CO$_2$ at 37 °C. The KO(Mm) and KO(Hs) cells with GFP and NLS-β actin re-introduced into KO cells were generated by retroviral transduction as described in[2].

**ATAC-seq.** Two biological replicates were used for each condition. Samples with 50,000 cells per condition were shipped in frozen medium (DMEM with 50 % FBS and 10% DMSO) on dry ice to Novogene (Beijing, China). All subsequent processing was performed by Novogene using standard DNA extraction and library preparation protocols. Cell nuclei were isolated, mixed with Tn5 Transposase with two adapters, and tagmentation was performed for 30 min at 37 °C. The fragmented DNA was purified and amplified with a limited PCR cycle using index primers. Libraries were prepared according to recommended Illumina NovaSeq6000 protocols. All ATAC-Seq processing was performed by Novogene (Beijing, China).

**ChIP-seq.** ChIP-Seq for BRG1, H3K9me3, and H3K27me3 was performed as described in ref.[2]. For EZH2 ChIP-Seq, Mouse Embryonic fibroblasts (two biological replicates per condition and input controls) were crosslinked using 1% formaldehyde (Sigma Cat. No. F8775) for 10 min followed by quenching with 0.125 M Glycine for 5 min and lysis with lysis buffer 1-LB1-(50 mM Hepes KOH pH 7.5, 10 mM NaCl, 1 mM EDTA, 10% glycerol, 0.5% NP-40, 0.25% Triton X-100). Nuclei were pelleted, collected and washed using lysis buffer 2-LB2- (10 mM Tris-HCl pH 8, 200 mM NaCl, 1 mM EDTA, 0.5 mM EGTA). This was followed by lysis using lysis buffer 3 LB3 (10 mM Tris-HCl pH 8; 100 mM NaCl, 1 mM EDTA; 0.5 mM EGTA; 0.1% Na-Deoxycholate, 0.5% N-laurylsarcosine). Chromatin was sheared using Qsonica Sonicator (4 cycles of 3 min at 70 % Amplitude), and then checked on 0.8% agarose gel. 100 μg of fragmented chromatin was mixed with 5 μg of (Ezh2 (D2C9) XP Rabbit mAb antibody-Cell signaling). The protein-antibody immunocomplexes were recovered by the Pierce Protein A/G Magnetic Beads (Thermo-Scientific). Beads and attached immunocomplexes were washed twice using Low salt wash buffer (LS) (0.1% SDS; 2 mM EDTA, 1% Triton X-100, 20 mM Tris-HCl pH 8, 150 mM NaCl), and High Salt (HS) wash buffer (0.1% SDS, 2 mM EDTA, 1% Triton X-100, 20 mM Tris-HCl pH 8, 500 mM NaCl), respectively. The beads were resuspended in elution buffer (50 mM Tris-HCl pH 8, 10 mM EDTA, 1% SDS). De-crosslinking was achieved by adding 8 μL 5 M NaCl and incubating at 65 °C overnight. RNase A (1 μL 10 mg/mL) was added for a 30 min incubation at 37 °C. Then, 4 μL 0.5 M EDTA, 8 μL 1 M Tris-HCl, and 1 μL 20 mg/mL proteinase K (0.2 mg/mL) were added for a 2-hour incubation at 42 °C to digest the chromatin. DNA was purified by QIAquick PCR purification kit (Qiagen, Germantown, MD, USA) for qPCR analysis and sequencing. ChIP-seq library preparation was done using the TruSeq Nano DNA Library Prep Kit (Illumina, San Diego, CA, USA) and then sequenced with the NextSeq550 sequencing platform (performed at the NYUAD Sequencing Center).

**HiC-seq.** Two biological replicates were used for each condition. Samples with 1 million cells per condition were fixed with 2% formaldehyde for 10 mins. The cell

pellets were washed twice by 1× PBS and then stored at −80 °C. Frozen pellets were shipped on dry ice to Genome Technology Center at NYU Langone Health, NY. All subsequent processing was performed by Genome Technology Center at NYU Langone Health using standard DNA extraction and library preparation protocols. Hi-C was performed by Genome Technology Center at NYU Langone Health, NY from 1 million cells. Experiments were performed in duplicates following the instructions from Arima Hi-C kit (Arima Genomics, San Diego, CA). Subsequently, Illumina-compatible sequencing libraries were prepared by using a modified version of KAPA HyperPrep library kit (KAPA BioSystems, Willmington, MA). Quality check steps were performed to assess the fraction of proximally ligated DNA labeled with biotin, and the optimal number of PCR reactions needed to make libraries. The libraries were loaded into an Illumina flowcell (Illumina, San Diego, CA) on a NovaSeq instrument for paired-end 50 reads.

**Co-immunoprecipitation and western blot**

*Nuclear and cytosolic fraction.* 10 million WT and KO cells were collected for nuclear and cytoplasmic fraction using Nuclear & Cytoplasmic Extraction Kit (G-Bioscience, 786-182, St. Louis, USA). Briefly, cells were trypsinized and counted by hemocytometer and the collected cell pellets were resuspended in 1000 μL of the cold SubCell Lysis buffer-I (from Kit) supplemented with Protease Inhibitor Cocktail (Roche, 11873580001, Basel, Switzerland). The cell suspension was immediately vortexed for 15 s at maximum setting and incubate on ice for 10 min. Then 50 μL SubCell Lysis Reagent (from Kit) was added to the cell suspension with immediate vortex. Upon incubation on ice for 5 min, the lysed cells were centrifuged for 5 min at maximum speed (~16,000×g). The supernatant containing the cytosolic fraction was transferred to a clean microtube on ice, and the nuclear pellet was resuspended in 200 μL Nuclear Extraction Buffer (from kit). After vigorously vortexing for 15 s, the microtube was incubated on ice for 30 min, with a vortex every 10 min. The nuclear extraction was then centrifuged for 10 min at maximum speed. Immediately, the supernatant containing soluble nuclear fraction was transferred to a new microtube on ice.

*Co-immunoprecipitation.* The soluble nuclear extraction was diluted in 800 μL SubCell Lysis buffer-I (from Kit) supplemented with Protease Inhibitor Cocktail. 100 μL cytoplasmic fraction and diluted nuclear fraction were transferred to new microtubes, snap-frozen on dry ice and stored at −80 °C as the input samples. In the remaining cytoplasmic and nuclear fractions, 6 μL of monoclonal anti-β-Actin antibody (Sigma, A2228, Taufkirchen, Germany) was added for each IP reaction and the microtubes were rotated in a shaker at 4 °C overnight. Then 30 μL Pierce Protein A/G Magnetic Beads (ThermoFisher Scientific, 88803, Waltham, USA) were added into each IP reaction and the microtubes were rotated in a shaker at 4 °C for 2 h. The microtubes were then placed on a magnetic separator to capture magnetic beads and the associated proteins. The supernatants were removed and the magnetic beads were washed 5 times with 1 mL washing buffer (20 mM Tris-HCl pH 7.4, 0.4 M NaCl, 1 mM EDTA, 0.5% Triton X-100, 10% glycerol, Roche Complete Protease Inhibitor Cocktail). The proteins precipitated were eluted in 150 μL 1× SDS Sample loading buffer (Sigma, S3401, Taufkirchen, Germany), and the input samples were thawed on ice and mixed with 100 μL 2× SDS Sample loading buffer. The protein samples were heated at 95 °C for 5 min and centrifuged at 2000×g for 1 min before gel loading.

*Western blot.* 15 μL protein sample was loaded into 10% polyacrylamide gel. Electrophoresis and transfer were performed using Trans-Blot® SD Semi-Dry Transfer Cell (Bio-Rad, Hercules, USA). The membrane was blocked for 1 h with 5% BSA in TBST buffer (Tris: 20 mM, NaCl: 150 mM, Tween® 20 detergent: 0.1% (w/v)). Immunoblot analysis was conducted with the following antibodies: β-Actin (Sigma, A2228, 1:1000); H3K27Me3 (Abcam, ab6002, 1:1000); Histone 3 (Abcam, ab1791, 1:1000); anti-Brg1 (Abcam, ab4081, 1:1000); Ezh2 (Cell Signaling, D2C9 #5246, 1:1000); REST/NRSF (Millipore, 17-641, 1:1000); GAPDH-HRP (Prosci, HRP-60004, 1:1000); goat anti-mouse IgG HRP (ThermoFisher Scientific, 62-6520, 1:2500), and goat anti-rabbit IgG HRP (ThermoFisher Scientific, 65-6120, 1:2500). Protein bands were developed with Pierce™ ECL Western Blotting Substrate (ThermoFisher Scientific, 32209) and were imaged using the ChemiDoc MP Imaging system (Bio-Rad, Hercules, CA, USA). Relative band densities were analyzed using ImageJ.

*Quantitative real-time PCR*. For qPCR of ACTA2, FLT1, and ITGB8, total RNA was isolated using TRIzol chloroform extraction according to the manufacturer's instructions. 200 ng to 1 µg total RNA was reverse transcribed to cDNA using quantitect reverse transcription kit (Qiagen). Diluted cDNA was subjected to quantitative real-time PCR analysis using Maxima SYBR Green qPCR Master Mix (ThermoFisher Scientific). All expression levels of target genes were normalized to the expression of Nono reference gene. qPCR analysis of DMP1, SPP1, and SDHA was performed as previously described[40,55]. The primers used for qPCR analysis are included in supplementary table 2

### Computational analysis

*RNA-seq*. RNA-Seq analysis was performed using published data for WT, KO, and HET mouse embryonic fibroblasts as described in ref. [2]. Raw counts for WT, β-actin KO and β-HET cells were downloaded from GSE95830 and DESeq2[56] was used to perform pairwise differential expression comparisons between WT/KO and WT/HET cells. Log2 fold change for genes overlapping different ATAC-Seq clusters and HiC compartments was averaged to generate Figs. 1H and 3F.

For repeat element analysis, repeat element annotation for the mouse genome was downloaded from the repeatmasker website (www.repeatmasker.org/genomes/mm10/RepeatMasker-rm405-db20140131/mm10.fa.out.gz) and filtered to exclude simple and low complexity repeats. RNA-Seq data were aligned to the genome using bowtie2 and analysis of repeat elements was performed using the Repenrich2[57] pipeline using default settings followed by a differential analysis of the resulting fraction_counts.txt files using edgeR[58].

*HiC-seq*. Raw sequencing data were processed using the HiCUP[59] pipeline (https://www.bioinformatics.babraham.ac.uk/projects/hicup/read_the_docs/html/index.html) and analyzed using HOMER[60] (http://homer.ucsd.edu/homer/) and HiC explorer[61,62] (https://hicexplorer.readthedocs.io/en/latest/). For preprocessing with HiCUP, digest file compatible with the Arima protocol was produced with HiCUP digester using the option –arima. Processed bam files produced by HiCUP were converted to HOMER format using the script hicup2homer followed by conversion to homer tag directories using the command makeTagDirectory -format HiCsummary. PCA analysis was performed using HOMER with the command runHiCpca.pl -genome mm10 -res 500000 -window 500000 followed by annotation and differential analysis with the scripts annotatePeaks.pl and getDiffExpression.pl. Bins changing PC1 values from positive to negative or vice versa with an FDR of less than 0.05 between WT and KO cells were classified as switching. Same analysis was also repeated using bin/window size of 50, 100, and 250 kb. Spearman correlation heatmap of 500 kb matrices was generated using the HiC explorer command hicCorrelate -method = spearman –log1p. Protein-coding genes overlapping with each compartment were identified using bedtools intersect with minimum overlap -f 0.51. RPKM normalized ChIP/ATAC-Seq counts associated with different compartments used in Fig. 5A were extracted from the relevant replicate merged bigwig tracks using deeptools function multibigwig summary. Normalized ChIP/ATAC counts associated with 100 kb bins in WT and KO cells used in Fig. 5B were extracted using the HOMER command annotatePeaks.pl with the option -rlog to apply variance stabilizing transformation (VST). GC and CpG content for different HiC compartments was obtained using the HOMER command annotatePeaks.pl -CpG. TAD analysis was performed with the HiCexplorer pipeline. To identify a consensus set of TAD boundaries, all tag directories for WT replicates were merged and converted to a hic file with the script tagDir2hicFile.pl. Merged hic file was converted to 50 kb matrix in cool format using the command hicConvertFormat followed by correction using hicCorrectMatrix with Knight-Ruiz algorithm. TADs were called using the command hicFindTADs using the options --correctForMultipleTesting fdr, --threshold 0.001, and --minBoundaryDistance 250000 –delta 0.01. To generate insulation score profiles, the same procedure was repeated for all samples with an additional normalization step using the command hicNormalize –smallest. Resulting bedgraph files were converted to bigwig format and insulation score at consensus WT boundaries was extracted for all samples. TAD boundaries overlapping different compartments were identified using bedtools[63] intersect with default parameters.

Significant Hi-C interactions were identified with the HOMER command findHiCInteractionsByChr.pl using a resolution of 25 kb and window size of 20 kb on replicate merged samples. To identify differential interactions the command was run first on the WT sample with KO as background (using the option -ped) and then on the KO samples with WT as background. All interactions were then annotated using the HOMER command annotateInteractions.pl (resulting files are included in extended data). Significant differential interactions overlapping genes were selected for plotting using manual analysis based on *p*-value and *z*-score differences. Matrices for 2 mb regions surrounding the selected interacting regions were generated using the HOMER command analyzeHic with the options -res 4000 -window 5000 -norm -nolog and plotted using the R package ggplot2. Plots of interaction frequencies as function of distance were generated for WT and KO cells by plotting the HOMER background model of replicate merged samples at 25 kb resolution.

*ATAC-seq and ChIP-Seq (EZH2) preprocessing*. Raw reads were quality trimmed using Trimmomatic[64] and analyzed with FastQC (http://www.bioinformatics.babraham.ac.uk/projects/fastqc) to trim low-quality bases, systematic base calling errors, and sequencing adapter contamination. Specific parameters used were

"trimmomatic_adapter.fa:2:30:10 TRAILING:3 LEADING:3 SLIDINGWINDOW:4:15 MINLEN:36". Surviving paired reads were then aligned against the mouse reference genome (GRCm38) using Burrows-Wheeler Aligner BWA-MEM[65]. The resulting BAM alignments were cleaned, sorted, and deduplicated (PCR and Optical duplicates) with PICARD tools (http://broadinstitute.github.io/picard). Bigwig files were generated using deeptools[66] command bamCoverage -bs 10 -e --ignoreDuplicates –normalizeUsingRPKM. Encode blacklisted regions were removed and replicate bigwig files were averaged using the deeptools[66] command bigwigCompare --operation mean. Bigwig files were analyzed with computeMatrix function of deeptools[66] to plot average signal around regions of interest.

*ATAC-seq differential analysis*. Differential analysis was performed using HOMER[60]. Processed bam files were converted to HOMER tag directories followed by annotation and differential analysis with the scripts annotatePeaks.pl and getDiffExpression.pl. ATAC-Seq peaks were called on cleaned, deduplicated bam files using macs2 with the parameters -q 0.05 -g mm --keep-dup all --nomodel --shift −100 --extsize 200 -B --broad -f BAMPE[67]. Peaks common to two replicates in each condition were retained and merged using homer command mergePeaks. Differential peaks were identified and annotated using homer scripts annotatePeaks.pl and getDiffExpression.pl. Peaks showing more than twofold change were divided into 3 clusters containing 84, 391 and 539 peaks using deeptools kmeans clustering. Clusters containing 391 and 539 peaks were classified as activated and repressed and used for further analysis. Promoters showing more than twofold change in ATAC signal with FDR < 0.05 were identified using HOMER command annotatePeaks.pl tss mm10 -size -1000,100 -raw and getDiffExpression.pl. To identify enhancers showing more than twofold change in ATAC signal with FDR < 0.05, enhancer-promoter pairs for MEFs were downloaded from http://chromosome.sdsc.edu/mouse/download/Ren_supplementary_table7.xlsx[18]. Regions 2 kb upstream and downstream of enhancer peaks were analyzed using annotatePeaks.pl and getDiffExpression.pl. To make plots of nucleosome occupancy, bam files were filtered to remove insert sizes <30 bp and processed with the NucleoAtac[68] pipeline using default settings and previously called atac-seq peaks. The resulting bedgraph files were converted to bigwigs and nucleosomal signal was plotted in 5 kb regions surrounding TSSs of interest using deeptools.

*EZH2 differential analysis*. Differential analysis was performed using HOMER. Processed bam files were converted to HOMER tag directories followed by annotation and differential analysis with the scripts annotatePeaks.pl and getDiffExpression.pl. Peaks were called using the macs2[67] command: macs2 callpeak -t Rep-1.bam Rep-2.bam -q 0.05 -g mm -c Input.bam (bam file for the relevant input control) --keep-dup all -B -f BAMPE --nolambda --nomodel –broad. WT and KO peaks were merged using HOMER command mergePeaks. Differential peaks were identified and annotated using homer scripts annotatePeaks.pl and getDiffExpression.pl. Overlap of TSSs with known polycomb targets for Fig. 2D was estimated using data from[19]. A transcript was regarded as a target if at least one polycomb subunit bound within 1 kb of its TSS. GC and CpG content for EZH2 peaks was obtained using the HOMER command annotatePeaks.pl -CpG

*BRG1, H3K9me3, H3K27me3, REST, and CTCF ChIP-Seq analysis*. For BRG1, H3K9me3, and H3K27me3 analysis, bigwig files were downloaded from GEO accession number GSE100096. Blacklisted regions were removed and replicates were averaged using the deeptools[66] bigwigCompare --operation mean. Merged and cleaned bigwig files were plotted using the deeptools commands computeMatrix, plotProfile, and plotHeatmap. Wig files for REST ChIP-Seq were downloaded from GEO accession GSM3331473 and GSM3331474. Bigwig file for mouse CTCF ChIP-Seq data was downloaded from the encode project https://www.encodeproject.org/files/ENCFF432JCZ/@@download/ENCFF432JCZ.bigWig.

Heatmaps showing pairwise Spearman correlation between various epigenetic marks in Fig. 5 and S4B were generated with deeptools[66] using replicate merged BAM files. To generate compartment-wise heatmaps, bed files of previously identified compartments at 500 kb resolution were divided in 5 kb bins using the bedtools[63] command makewindows and supplied to deeptools command multiBamSummary and plotCorrelation with default settings.

*GO term analysis*. All GO term analyses were performed using Metascape[69] https://metascape.org/.

### Statistical tests

**Statistical tests**. Statistical significance for all figures was calculated using GraphPad Prism for Windows, GraphPad Software, La Jolla California USA, www.graphpad.com, and R packages rstatix and ggpubr.

**Reporting summary**. Further information on research design is available in the Nature Research Reporting Summary linked to this article.

## Data availability
The data that support this study are available from the corresponding author upon reasonable request. Next-generation sequencing data reported in this study have been deposited in the Gene Expression Omnibus with accession numbers GSE149987 and

GSE133196. Raw counts for RNA-seq analysis of WT, β-actin KO, and β-HET cells were downloaded from GSE95830. For BRG1, H3K9me3, and H3K27me3 ChIP-seq analysis, bigwig files were downloaded from GEO accession number GSE100096. Wig files for REST ChIP-Seq were downloaded from GEO accessions GSM3331473 and GSM3331474. Bigwig file for mouse CTCF ChIP-Seq data was downloaded from the encode project with accession ENCFF432JCZ. Source data are provided with this paper.

## Code availability

All analysis was performed using commercial or publicly available software packages. The details of the software and parameters used for various analyses are provided in the relevant "Methods" section. Scripts used for generating specific figures are available upon request.

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

## Acknowledgements
We thank Nizar Drou, Marc Arnoux, and Mehar Sultana from the genomics core facilities at NYU Abu Dhabi, Center for Genomics and Systems Biology for technical help. We thank Christophe Ampe (University of Gent, Belgium) for kindly providing us with the β-actin +/+, β-actin +/−, and β-actin −/− MEFs. This work is supported by grants from New York University Abu Dhabi, the Sheikh Hamdan Bin Rashid Al Maktoum Award for Medical Sciences, the Swedish Research Council (Vetenskapsrådet), and the Swedish Cancer Society (Cancerfonden).

## Author contributions
X.X. and P.P. conceptualized the research. S.R.M., X.X., N.H.E.S., and T.V. conducted the experiments. S.R.M. performed all computational analysis and made the figures. S.R.M. and P.P. wrote the manuscript. P.P. and K.C.G. supervised the research. All authors read and approved the manuscript.

## Competing interests
The authors declare no competing interests.
