## [Peer Review File · Nature Communications]

β -actin dependent chromatin remodeling mediates compartment level changes in 3D genome architectureEditorial Note: Parts of this Peer Review File have been redacted as indicated to remove third-party material where no permission to publish could be obtained.

REVIEWER COMMENTS

Reviewer #1 (Remarks to the Author):

In this manuscript, Syed Raza Mahmood et al, explore the role of nuclear β -actin in chromatin accessibility, histone modifications and 3D genome architecture. The authors use as a tool mouse embryonic fibroblast (MEFs) knock-out for β -actin. The work is interesting, although some additional experiments are needed to support their claims.

General comment: Loss of β -actin is compensated by the expression of other family members. Is this compensation also happening in the nucleus? Or is specific to maintain basic cytoskeletal roles?

1. Figure 1A: the authors claim that β -actin expression has been analysed at RNA and protein level. I cannot find the mentioned western blot on the manuscript. Moreover, a qRT-PCR is also needed, RNA data from the sequencing is not enough. What happens with the expression of other family members?
2. The authors use several times the word "repression" or "activation" when they refer to less or more accessible chromatin. It has been reported that an open chromatin does not mean active chromatin and the other way around. There are many examples in where we can observe more accessibility, but this does not always correlate with active transcription. Authors should change these words for less or more accessible chromatin or less or more ATAC signal.
3. The authors have analysed through the manuscript the expression levels of genes located in their areas of interest, which is fine. However, since one of the effects that the authors observe is heterochromatin opening, analysing the expression of major satellites will be very nice.
4. In cluster 1 and 2, higher levels of EZH2 and H3K27me3 are observed, although the distribution of the histone mark is quite different. We can observe a sharp peak or a more broader profile. Could the authors discuss a bit this observation? In the literature, there are many examples with this different pattern, a discussion on that will increase the quality of the conclusions. The authors have to show the global levels of H3K27me3 and H3K9me3 by western blot in WT vs KO.
5. Authors suggest that in the absence of actin, BRG1-BAF complex are released and chromatin gets accessible. BRG1 ChIPseq should be done to claim this. Or at least select several regions and show by ChIP-qPCR the BRG1 release. Also in these regions, the complex is acting as a repressor. The authors suggest that this could happen when it interacts with NurD or REST. Chip-seq of NurD and REST would be essential.
6. Why the correlation between EZH2 and H3K27me3 is too low?
7. Figure 2E. In order to claim that the effect is dose dependent, the authors should show the Het for all the panels, not only EZH2.
8. Why KO-mouse/human actin cells do not cluster with the WT? Rescue cells also rescue epigenetic changes observed in KO?

Reviewer #2 (Remarks to the Author):

The authors conducted a detailed study of BRG1 and EZH2 interplay on genome organization after β -actin knock out. A number of assays, including Hi-C is used to analyze genome changes. The authors report that β -actin regulates a portion of A and B compartments while not affecting the insulation score (loop structure). Although the results shown are largely clear, some conclusions are not convincing.

Major points:

1. β -actin dosage-dependent effect on genome organization is not convincing. Although the authors showed that cells heterozygote (HET) for β -actin have moderate effects comparing to KO, the rescue effect of nuclear localized mouse or human β -actin is subtle. To further support the notion of dosage-dependent effect of β -actin, the authors should explore the effect of β -actin overexpression. The authors also didn't show the level of β -actin protein in different samples. It is hard to make any conclusion just based on mRNA level.
2. The authors state the nuclear β -actin plays a major role for genome architecture change in the β -actin KO cells. However, they did not show sufficient evidence to rule out the possibility that β -

actin may function in the cytoplasm. It is important to compare the nuclear β -actin protein level to that of the cytoplasm level. Could cytoplasm β -actin affect some other protein which in turn indirectly affect nuclear BRG1 or EZH2? It is also confusing why expressing β -actin did not fully rescue genome organization as shown in Fig. 3E.

3. The authors states " β -actin KO cells exhibit close to zero β -actin expression both at the mRNA and protein level while HET cells show expression levels intermediate between WT and KO cells (8) (Figure 1A)." The protein level information was not present in the manuscript. And the figure in the reference cited indicate that the β -actin level in the HET is comparable to that of the WT cells.

4. Fig 1C and 1D do not appear to agree with one another well. The total down-regulated peak number is 575, and only 28% (168) of them overlapped with TSS (Fig 1C). In Fig 1D, however, there are about 300 TSS showed decreased ATAC signal.

5. Although studying BRG1 and EZH2 occupancy in the genome represent a large portion of the paper, only RNA level of these proteins is provided. It is important to show that BRG1 and EZH2 protein levels are similar or different in KO, HET and wild type cells.

6. In the compartment analysis, there's some A to B or B to A switch after β -actin KO. How consistent are these switches? What kind of variation between different samples in the same group? In fig 3D, for the converted compartment, how many bins overlap between rescue experiments using mouse or human β -actin? does it mean actin only function on a small population of compartment? What are the genes related with the common rescued compartment?

7. Page 6, "Switching compartments also showed expected changes in the transcriptional (Figure 4A)". This statement is unclear based on the data. There are some genes showing increased gene expression even though they switched from A compartment to B compartment. More importantly, in the B to A switched genes, half of them showed increased gene expression. Please explain why after compartment switch, some show gene expression change while others do not. Please also provide the information for X axis. Are they replicates?

8. Fig S6A, although Sox11 and Zfp57 genes showed compartment B to A change, they didn't show increased genome accessibility.

9. The correlation analysis of epigenetic marks in Fig 5A is confusing. Better explanation is needed so the readers can understand the correlation changes between WT and KO. For example, if BRG1 binding suppresses the binding by EZH2 or modification by H3K27Me3, then we should expect negative correlations in both A and B compartments in the absence of β -actin. But the correlations shown are positive.

Minor points:

1. Page 4, statement: "BRG1, EZH2, and H3K27me3 were much more highly correlated with each other in KO cells as compared to WT cells (Figure S4B)". There is no explanation about why BRG1, EZH2, and H3K27me3 should correlate with each other. What's the purpose of this analysis?

2. Page 5, statement: "Similar to our observations for sites showing increased accessibility in KO cells, GO term analysis of regions gaining EZH2 peaks revealed an enrichment of biological processes linked to skeletal system and vasculature development (Figure 2F and Figure 1E)". Besides GO term, please also provide the list of genes showing overlap between EZH2 peaks and ATAC-seq peaks.

3. Page 5, there's no figure S3B and S3D. Please clarify the labelling.

4. As Hi-C resolution highly depends on the quality of the data and the author try to use different resolution to analysis the data, please provide a summary of the Hi-C data, such as raw read pair number, valid pair number, Intra-chromosome read pair number and Inter-chromosome read pair number.

5. Page 6, statement: "As compartment identity is reflected in the sign of the first principal component (PC1)," Please provide the reference that support this statement.

6. The figure order is not well organized. I had to jump from different figures when I read the paper.

7. Page 7, statement: "TAD analysis at 50kb resolution revealed that a majority of TAD boundaries showed little or no difference in insulation scores between WT and KO cells (Figure 3G and Figure 4A)." Fig 4A is totally unrelated to insulation scores.

8. Page 10, statement: "Since H3K9me3 is characteristic of HP1 α positive heterochromatin and shows a strong correlation with compartment switching in β -actin KO cells.". Since the R2 is only 0.37, I am not sure why the authors believe it is a strong correlation.

9. Methods, no input information is provided for ATAC-Seq and ChIP-seq.
10. Figure S10 is not mentioned in the main text.
11. The number of replicates for each experiment needs to be mentioned.
12. Gene list for each GO term needs to be provided.

Reviewer #3 (Remarks to the Author):

The relatively recent discovery of actin in the nucleus opens a very exciting field of research. Furthermore the hierarchical folding of genome folding is central to gene expression, cellular development and function. The submitted manuscript by Syed Raza Mahmoud and colleagues is in these lines of research, to understand the function of β -actin in chromatin remodeling and the 3D structure of the genome. Using a battery of genome-wide techniques including ATAC-seq, Chip-seq, RNA-seq and HiC, in mouse embryonic fibroblast (MEF) cells completely or partially deleted for β -actin, the authors find that changes in actin levels are associated with different chromatin accessibility and changes in the 3D structure of the genome. The authors propose a new role for β -actin in reshaping the genomic landscape that could be important in the regulation of genes involved in differentiation and development.

The exploitation of genome-wide data and their smart analyses are clearly assets for a global understanding of the modifications induced by changes in actin levels. However, in my opinion, the conclusions should be supported as much as possible by orthogonal approaches. This would give a definitely more solid character to the final message.

To begin with, the authors work on β -actin $-/-$ or β -actin $-/+$ heterozygous MEFs whose actin level of expression is verified by RNA seq. It would be good for the authors to i) specify how the KO was constructed and verified, ii) confirm RNA-seq result by RT-QPCR and iii) provide a WB in order to confirm the intermediate level of β -actin in heterozygous cells (not shown in Tondeleir et al., 2012).

The group has already shown in a previous study that the KO of β -actin has an effect on the nuclear position of the epigenetic heterochromatin constitutive mark H3K9me3 (by IF), that the level of this mark varies (by ChIP-seq) and that these effects are accompanied by an increase or decrease in various genes expression (Xie et al., 2018). In the present study the authors further correlate these changes in expression with differences in chromatin accessibility (ATAC-seq). It would be useful to clarify whether the up and down regulated genes are the same in both studies and also to confirm the absence of cleavage bias by Tn5 by testing, for example, the positioning of nucleosomes for a few positions in the WT and KO strain.

It is interesting that among the regions that have lost accessibility some correspond to TSS generally enriched with chromatin remodelers such as BAF/BRG1. The authors have already shown by ChIP-seq that the KO of β -actin leads to a loss of BRG1 on chromatin (Xie et al., 2018). This supposes a direct interaction between BRG1 and β -actin that would benefit from being shown by co-IP in the present study. Why is ChIP-seq in heterozygous cells not shown for BRG1? (Figure 2A). More generally, can the authors comment on the results obtained by Barisic et al (2019) and their own (i.e. deletion of BRG1 versus β -actin KO)?

The authors went further in the regulatory role of actin in the chromatin configuration by making HiC not only in KO and HET, but also in add-back experiments. The authors observed global changes in the interactions between genomic regions without changing the nature of compartments A (euchromatin) and B (heterochromatin), except for 6% of the genome, which switch between the two compartments.

It would definitely be necessary to confirm by DNA FISH the switch between compartments A and B by measuring the distances between a few pairs of genes in a given compartment and comparing them between WT and KO cells.

As expected the levels of expression by RNAseq confirms the transition from compartment A to

compartment B (and vice-versa). It would be useful for the authors to confirm this by showing an RT-QPCR of selected genes.

Given the HiC results, it could be interesting to calculate the probability of contact as a function of genomic distance $P(s)$ in order to define the potential effects of actin loss on the chromatin fibre structure. I also advise to make a differential analysis of the contacts between WT and KO, to quantify them more precisely (for instance see the star method from Kundu et al., 2017, Mol Cell).

Because the effect of actin on chromatin is different according to the cell cycle, it would be interesting in the future to refine this analysis on synchronized cells.

Rebuttal letter – manuscript no. NCOMMS-20-25611-T

We would like to thank all reviewers for their constructive feedback. Please find below our point-by-point responses (black text) to each of the reviewers' concerns (red text).

Reviewer #1:

In this manuscript, Syed Raza Mahmood et al., explore the role of nuclear β -actin in chromatin accessibility, histone modifications and 3D genome architecture. The authors use as a tool mouse embryonic fibroblast (MEFs) knock-out for β -actin. The work is interesting, although some additional experiments are needed to support their claims.

We thank the reviewer for the helpful feedback. We have included additional experiments and analyses in the revised manuscript to address the reviewer's concerns.

General comment: Loss of β -actin is compensated by the expression of other family members. Is this compensation also happening in the nucleus? Or is specific to maintain basic cytoskeletal roles?

While β -actin KO cells do exhibit upregulation of other actin isoforms and compensation of cytoskeletal functions such as migratory capacity (Tondeleir et al., 2012), it is unlikely that such compensation occurs in the cell nucleus. So far there is no evidence supporting the presence of alpha actin in the cell nucleus. Furthermore, this study focuses only on the role of nuclear actin as a component of the BAF complex subunit BRG1. Since the BAF complex specifically contains the β -actin isoform, it is unlikely that the upregulation of other isoforms of actin would be able to rescue BRG1 function. Our data also supports this idea as β -actin KO cells show almost complete loss of BRG1 chromatin association despite the upregulation of other actin isoforms (see Xie et al 2018). In contrast, it is likely that in the β -actin KO background, other actin isoforms (in particular alpha actin) may compensate for the role of β -actin in the cytoplasm since the cells display an intact cytoskeleton (see Xie et al., 2018)

1. Figure 1A: the authors claim that b-actin expression has been analysed at RNA and protein level. I cannot find the mentioned western blot on the manuscript. Moreover, a qRT-PCR is also needed, RNA data from the sequencing is not enough. What happens with the expression of other family members?

In this study we have reanalyzed published RNA-Seq data from our previous work which included both qRT-PCR validation of RNA-Seq results and immunoblots of β -actin protein levels (Xie et al., 2018). For the reviewer's reference, we are reproducing below the relevant RT-qPCR data and immunoblots from that study (Xie et al., 2018). The RT-qPCR data shows β -actin expression in WT, KO and HET cells as well as the upregulation of α -actin and γ -actin upon β -actin loss. The immunoblot shows the expression of β -actin in WT, KO and HET cells.

[REDACTED]

qPCR analysis of actin isoforms in WT,
KO and HET cells normalized to Nono
Figure-8, Xie et al. ,2018

[REDACTED]

Western blot analysis of β -actin in
WT, KO and HET cells normalized to
GAPDH

Figure-8, Xie et al. ,2018

2. The authors use several times the word “repression” or “activation” when they refer to less or more accessible chromatin. It has been reported that an open chromatin does not mean active chromatin and the other way around. There are many examples in where we can observe more accessibility, but this does not always correlate with active transcription. Authors should change these words for less or more accessible chromatin or less or more ATAC signal.

We thank the reviewer for pointing this out. We have replaced ‘repression’ and ‘activation’ with ‘less accessible’ and ‘more accessible’ when referring to changes in accessibility.

3. The authors have analysed through the manuscript the expression levels of genes located in their areas of interest, which is fine. However, since one of the effects that the authors observe is heterochromatin opening, analysing the expression of major satellites will be very nice.

We thank the reviewer for this suggestion. We have now reanalyzed our RNA-Seq data using the Reperich2 pipeline which allows differential analysis of various repeat families including transposons and satellites. An overview of the repeat element classes showing differential expression between β -actin WT and KO cells is now included in supplementary figure S12. Our data shows minimal changes in the expression of satellites but reveals downregulation of several LINE and DNA repeat families. Such downregulation is consistent with the increase in H3K9Me3 and H3K27Me3 observed in β -actin KO cells. A more detailed analysis of non-coding and repetitive elements in β -actin KO cells will be a part of future studies from our lab.

4. In cluster 1 and 2, higher levels of EZH2 and H3K27me3 are observed, although the distribution of the histone mark is quite different. We can observe a sharp peak or a more broader profile. Could the authors discuss a lit bit this observation? In the literature, there are many examples with this different pattern, a discussion on that will increase the quality of the conclusions.

Results from our ChIPseq analysis demonstrate that β -actin depletion leads to increased levels of Ezh2 and H3K27me3 around TSSs. We believe that the slight differences in the peaks' shapes mentioned by this reviewer reflect actin-dependent alterations in Ezh2 and H3K27me3 occupancies which can have important effects on transcription. It has been shown, for instance, that H3K27me3 can be broadly enriched across gene bodies or be specifically enriched at TSSs of bivalent or active genes and these patterns can be important for transcriptional activity (Young et al., 2011). While a more detailed analysis of the shape of the profiles of specific epigenetic marks is interesting, we believe it is beyond the scope of the present study. An analysis of actin-dependent changes in methylation patterns of different groups of genes in WT and KO cells and how these patterns correlate with EZH2 occupancy is, nevertheless, an interesting area of future research.

The authors have to show the global levels of H3K27me3 and H3K9me3 by western blot in WT vs KO.

Actin-dependent H3K9me3 levels were investigated in a previous study from our lab (see Xie et al 2018) and are reproduced below for convenience. In the revised manuscript, we have now included

[REDACTED]

Immunoblots of H3K9me3 in WT, KO and HET cells normalized to total histone 3

Figure-8, Xie et al. ,2018

immunoblots of H3K27me3 in WT and KO cells (see supplementary figure S4C).

5. Authors suggest that in the absence of actin, BRG1-BAF complex are release and chromatin gets accessible. BRG1 ChIPseq should be done to claim this. Or at least select several regions and show by

ChIP-qPCR the BRG1 release. Also in these regions, the complex is acting as a repressor. The authors suggest that this could happen when interacts with NurD or REST. Chip-seq of NurD and REST would be essential.

The BRG1 ChIP-seq is already included in the manuscript and actin-dependent loss of BRG1 chromatin binding is clearly illustrated in Figure 2A and 2B (see also Xie et al 2018). To further support the claim that actin is required for Brg1 association we have now included a Co-IP of β -actin and BRG1 from fractionated cellular extracts (see Figure 1A). The results suggest a direct actin-Brg1 interaction that is likely to occur in the cell nucleus and has been recently substantiated by the crystal structure of the BAF complex (Mashtalir et al., 2020).

As for the co-repressors REST and NuRD, we do not claim that β -actin directly affects their association with chromatin, rather we believe that changes in NURD and REST occupancies are indirect effects of the actin-dependent loss of Brg1. In support of our reasoning, previous studies have already confirmed that loss of BRG1 function directly impacts the ability of REST to bind chromatin (Barisic et al., 2019; Matsumoto et al., 2006; Ooi et al., 2006). For this reason, we believe that a detailed analysis of REST and NuRD binding are beyond the scope of the present work as they constitute one of the many potential downstream effects of BRG1 loss.

However, as per the reviewer's suggestion, we have performed ChIP-Seq of REST in WT and KO cells and have included some preliminary results below. As expected, our results demonstrate significant loss of REST chromatin binding in β -actin KO cells. This result is consistent with Barisic et al., 2019, which similarly reported loss of REST chromatin association in BRG1 KO cells and confirms that β -actin mediated loss of BRG1 binding also impacts the ability of REST to bind chromatin. While our data shows identical REST binding between the differential ATAC-Seq clusters (Cluster 1-Activated and Cluster 2-Repressed), we believe that REST occupancy in differentiated mouse embryonic fibroblasts cannot reflect its relationship with BRG1 in stem cells and during differentiation. The REST ChIP-Seq data from Barisic et al., 2019 included in figure 2E vi, is therefore more appropriate for the present study since it reflects REST occupancy in mouse embryonic stem cells rather than in differentiated fibroblasts. Uncovering the relationship between REST and other remodelers in β -actin KO cells is a part of ongoing

Normalized ChIP-Seq signal (RPKM) at the promoters of all protein-coding genes showing significant loss of REST binding in β -actin KO cells

Normalized ChIP-Seq signal (RPKM) at the peak centers of Cluster 1 and Cluster 2 in WT cells. In comparison with mESC ChIP-Seq (Fig 2E vi), both clusters exhibit similar REST occupancy in MEFs

work in our lab.

6. Why the correlation between EZH2 and K27me3 is too low?

While the correlation between EZH2 and H3K27Me3 in WT cells is relatively low (0.29), it is significantly higher in KO cells at 0.58. We believe that it is difficult to obtain a perfect correlation between different epigenetic marks due to the different signal to noise ratios and immunoprecipitation efficiencies of the different antibodies. The relative difference in the correlation of different epigenetic marks between WT and KO cells as shown in Figure 5A is therefore more meaningful than the absolute correlation in one cell type. Another possible reason for the relatively low correlation could be the fact that the EZH2 containing PRC2 complex is not only associated with H3K27me3 but also catalyzes H3K27me1 and H3K27me2 which are broadly distributed over gene bodies and throughout the genome (Lavarone et al., 2019, p. 27). Since H3K27me3 is more specifically enriched at promoters, it would not correlate highly with EZH2 at non-promoter regions such as gene bodies.

7. Figure 2E. In order to claim that the effect is doses dependent, the authors should show the Het for all the panels, not only EZH2.

The main focus of the present study is to demonstrate the dependence of chromatin accessibility, compartment organization and transcription on β -actin levels as shown by ATAC-Seq (Figure 1G), HiC-Seq (Figure 3D) and RNA-Seq (Figure 1H and 3F). As all of these assays demonstrate an intermediate phenotype in the heterozygous cells, we believe that relationship between β -actin levels and chromatin structure is convincing. However, we agree with this reviewer that claiming direct dosage dependence between β -actin levels and specific epigenetic marks would need further investigation. We have, therefore, removed the EZH2 ChIP-Seq data for the heterozygous sample from Figure 2 and have also modified the text accordingly. The revised manuscript highlights the relationship between β -actin levels and chromatin structure but does not claim dosage dependence between β -actin levels and any specific epigenetic mark.

8. Why KO-mouse/human actin cells do not cluster with the WT? Rescue cells also rescue epigenetic changes observed in KO?

As we have demonstrated in the present study, loss of β -actin has a significant impact on the activities of diverse chromatin remodelers such as BRG1, EZH2 and REST. Since the primary function of these remodeling complexes is to regulate chromatin structure during differentiation, it is unlikely that the reintroduction of mouse/human β -actin, in fully differentiated embryonic fibroblasts would reverse all the changes that β -actin loss induced in the activities of these remodelers over the course of cell differentiation. The clustering together of rescue samples with KO cells reflects the fact that both these cells developed without β -actin from the earliest stages of differentiation and hence their chromatin structure is more similar to each other than to WT cells. Furthermore, the expression of NLS- β -actin in KO cells cannot restore β -actin levels inside the nucleus to the same level as WT cells (Xie et al., 2018). As a result, we don't expect full rescue of epigenetic changes at the genomic level upon reintroduction of NLS-actin. However, we have demonstrated β -actin mediated rescue of epigenetic marks at the level of specific genes like Cebpa in our recent work (Al-Sayegh et al., 2020).

Reviewer #2:

The authors conducted a detailed study of BRG1 and EZH2 interplay on genome organization after β -actin knock out. A number of assays, including Hi-C is used to analyze genome changes. The authors report that β -actin regulates a portion of A and B compartments while not affect the insulation score (loop structure). Although the results shown are largely clear, some conclusions are not convincing.

We thank the reviewer for the constructive feedback and have tried to address the concerns below.

Major points:

1. β -actin dosage-dependent effect on genome organization is not convincing. Although the authors showed that cells heterozygote (HET) for β -actin have moderate effects comparing to KO, the rescue effect of nuclear localized mouse or human β -actin is subtle. To further support the notion of dosage-dependent effect of β -actin, the authors should explore the effect of β -actin overexpression.

We agree with the reviewer that the rescue effect of NLS-mouse/human β -actin is subtle. However, it is important to note that the loss of β -actin has a significant impact on the activities of chromatin remodelers such as BRG1, EZH2 and REST whose primary function is to regulate chromatin structure during differentiation and development. It is therefore not surprising that the reintroduction of mouse/human β -actin in fully differentiated embryonic fibroblasts does not completely rescue the chromatin structure of these cells because β -actin dependent changes in the remodeling activities of these complexes would have occurred over the course of cell differentiation.

Furthermore, given the diverse roles of actin in both the nucleus and the cytoplasm, we believe that significant overexpression of β -actin can not only have unintended effects on cellular function but will also make it difficult to differentiate between the contribution of nuclear and cytoplasmic actin pools on any observed rescue. Similarly, it is unlikely that stronger overexpression of β -actin in differentiated fibroblasts would completely reverse any changes in β -actin-dependent chromatin remodeling that occurred over the course of differentiation. However, we believe that the dosage dependent changes in chromatin accessibility and compartment organization shown in heterozygous MEFs by ATAC-Seq and HiC-Seq provide convincing evidence for the role of β -actin levels in regulating chromatin structure as these MEFs developed with intermediate levels of β -actin from the earliest stages of development.

The authors also didn't show the level of β -actin protein in different samples. It is hard to make any conclusion just based on mRNA level.

We have demonstrated the levels of β -actin in WT, KO and HET cells by immunoblots in our previous work (Xie et al., 2018). We are reproducing the relevant immunoblot below for reference (see below).

[REDACTED]

Western blot analysis of β -actin in
WT, KO and HET cells normalized to
GAPDH

Figure-8, Xie et al. ,2018

2. The authors state the nuclear β -actin plays a major role for genome architecture change in the β -actin KO cells. However, they did not show sufficient evidence to rule of the possibility that β -actin may function in the cytoplasm. It is important to compare the nuclear β -actin protein level to that of the cytoplasm level. Could cytoplasm β -actin affect some other protein which in turn indirectly affect nuclear BRG1 or EZH2?

While we cannot completely discount the possibility that loss of cytoplasmic β -actin may somehow indirectly affect BRG1 via another protein, to the best of our knowledge no other studies have reported such a mechanism. Based on current evidence nuclear β -actin levels therefore seems to be the primary factor affecting BRG1 function. To further illustrate the direct interaction between β -actin and BRG1 inside the nucleus, we have now also performed a Co-IP of β -actin and BRG1 from nuclear and cytoplasmic fractions in WT and KO cells (Figure 1A). Our data clearly shows that BRG1 is not present in the cytoplasmic fraction and only co-immunoprecipitates with β -actin in the nuclear fraction of WT cells. This implies a direct interaction between β -actin and BRG1 in the nucleus and is consistent with other studies showing that the association of β -actin and BRG1 is essential to BAF function and is compatible with the recently solved crystal structure of the BAF complex (Mashtalir et al., 2020; Nishimoto et al., 2012; Zhao et al., 1998).

It is also confusing why expressing β -actin did not fully rescue genome organization as shown in Fig. 3E.

We agree with the reviewer that the rescue effect of NLS-mouse/human β -actin is partial. As discussed in the response the point no 1, this subtle rescue effect is consistent with the fact that the changes in genome organization mediated by chromatin remodelers like BRG1 and EZH2 occur during the process of cell differentiation and such changes can therefore not be fully reversed in differentiated cells. Furthermore, reintroduction of NLS- β -actin in KO cells cannot restore β -actin expression in the nucleus to the same level as WT cells and any rescue affect is therefore expected to be partial.

3. The authors states “ β -actin KO cells exhibit close to zero β -actin expression both at the mRNA and protein level while HET cells show expression levels intermediate between WT and KO cells (8) (Figure 1A).” The protein level information was not present in the manuscript. And the figure in the reference cited indicate that the β -actin level in the HET is comparable to that of the WT cells.

In addition to the western blot provide in response to point 1, , we are also reproducing below RT-qPCR

[REDACTED]

qPCR analysis of β -actin in WT, KO and HET cells normalized to Nono
Figure-8, Xie et al. ,2018

7

CPM normalized counts for the β -actin gene in WT, KO and HET cells
Figure 1B

and RNA-Seq data for β -actin (Xie et al., 2018). Together, these data show that heterozygous cells express β -actin at an intermediate level.

4. Fig 1C and 1D do not appear to agree with one another well. The total down-regulated peak number is 575, and only 28% (168) of them overlapped with TSS (Fig 1C). In Fig 1D, however, there are about 300 TSS showed decreased ATAC signal.

The difference between the two figures arises from the fact that Fig 1C is based on a differential analysis of ATAC-Seq peaks and hence only shows downregulated TSSs that are associated with a peak (28% or 168 peaks). In contrast, Fig 1D is based on a differential analysis of ATAC-Seq signal at all promoter regions irrespective of whether these regions constitute a peak or not. It hence shows a larger number of TSS sites (~300) that have decreased accessibility.

5. Although studying BRG1 and EZH2 occupancy in the genome represent a large portion of the paper, only RNA level of these proteins is provided. It is important to show that BRG1 and EZH2 protein levels are similar or different in KO, HET and wild type cells.

We have included immunoblots for BRG1 and EZH2 in supplementary figure S4B.

6. In the compartment analysis, there's some A to B or B to A switch after β -actin KO. How consistent are these switches? What kind of variation between different samples in the same group? In fig 3D, for the converted compartment, how many bins overlap between rescue experiments using mouse or human β -actin? does it mean actin only function on a small population of compartment? What are the genes related with the common rescued compartment?

To define A to B or B to A switching compartments, PC-1 values of two biological replicates for each 500kb bin were averaged and bins whose average PC-1 values changed from positive to negative or vice versa with a p-value less than 0.05 were classified as switching. The PC-1 values were highly consistent between the two replicates. We have included a heatmap showing PC-1 values for each replicate in supplementary figure S5B.

Out of the 168 bins that switched from B to A upon actin loss, 89 (52%) and 69 (41%) reverted back to their original B state upon reintroduction of mouse or human actin respectively (Figure 3D). Among these 57 rescued bins were common to both human and mouse rescue samples.

Out of the 125 bins that switched from A to B upon actin loss, only 23 (18%) and 16 (13%) reverted back to their original A state upon reintroduction of mouse or human actin respectively (Figure 3D). Of these 12 were common to both human and mouse rescue samples.

Lists of genes associated with bins rescued by both mouse/human actin are now provided in supplementary table S13.

While it is true that only a small percentage of switching bins were rescued, as previously mentioned the reintroduction of mouse/human β -actin into differentiated cells is unlikely to reverse all β -actin mediated changes that may have occurred during differentiation. It is therefore possible that β -actin levels influence a larger proportion of genomic regions during development than suggested by the rescue experiment in differentiated fibroblasts.

7. Page 6, “Switching compartments also showed expected changes in the transcriptional (Figure 4A)”. This statement is unclear based on the data. There are some genes showing increased gene expression even though they switched from A compartment to B compartment. More importantly, in the B to A switched genes, half of them showed increased gene expression. Please explain why after compartment switch, some show gene expression change while others do not. Please also provide the information for X axis. Are they replicates?

Compartments constitute genomic regions that have a high frequency of interactions within themselves and a low frequency of interactions with regions classified as the opposite compartment. Such A and B compartments are known to correlate with ‘active’ and ‘inactive’ genomic regions in terms of average expression. Consistent with this idea, our data shows that **on average**, A to B switching regions show a negative change in expression while B to A switching regions show a positive average change (Fig 3F). However, as these compartments represent large areas of the genome, the change in interaction frequencies that underlies a compartment switch will not always translate to a change in expression for every gene overlapping the switching region. This is especially true in the case of B to A switching genes as a change in chromatin state or interaction frequencies will not necessarily be sufficient for transcriptional activation in the absence of downstream factors such as transcription factor availability or other forms of transcriptional/post-transcriptional gene regulation. This idea ties in with the concept of activatable and occluded genes which respectively require either chromatin independent mechanisms like transcription factors or chromatin based derepression mechanisms to be properly activated (Miyamoto and Gurdon, 2013). As a result, we believe that the expression changes shown in Fig 4A are a reasonable representation of compartment switching where majority of the 100 most variable genes show downregulation after an A to B switch and a significant proportion show upregulation in response to a B to A switch. This is further confirmed by the average expression changes shown in figure 3F which includes data for all genes overlapping each compartment.

The x-axis of Figure 4A represents biological replicates. We have added this information to the figure.

8. Fig S6A, although Sox11 and Zfp57 genes showed compartment B to A change, they didn’t show increased genome accessibility.

As discussed in the response to point no. 7, a compartment switch represents a change in the frequency of interactions between different genomic regions and may or may not lead to changes in chromatin accessibility of specific features like promoters of individual genes. Changes in gene expression upon compartment switching are not always dependent on chromatin accessibility but can also be induced by chromatin independent modes of regulation such as changes in enhancer-promoter contacts, transcription factor availability or other forms of transcriptional/post-transcriptional regulation. Consistent with this idea our RNA-Seq data shows that despite not showing increased chromatin accessibility, Sox11 is significantly upregulated (Log2FC:1.23, FDR:3.04x10⁻⁷) upon switching from the B to A compartment. (Zfp57 also shows a positive Log2FC but does not satisfy the 5% FDR cutoff)

9. The correlation analysis of epigenetic marks in Fig 5A is confusing. Better explanation is needed so the readers can understand the correlation changes between WT and KO. For example, if BRG1 binding suppresses the binding by EZH2 or modification by H3K27Me3, then we should expect negative correlations in both A and B compartments in the absence of β-actin. But the correlations shown are positive.

BRG1 has a complex relationship with polycomb group proteins and H3K27me3. While it has been shown that BRG1 deposition can induce the displacement of polycomb proteins, it has also been shown that the BRG1 containing BAF complex co-localizes with polycomb proteins at up to 67% of its binding sites suggesting that the two complexes can also cooperate (Kadoch et al., 2017). Similarly, BRG1 has also been shown to be required for the deposition of H3K27me3 at certain developmental regulators (Alexander et al., 2015). The precise relationship between BRG1, EZH2 and H3K27me3 may therefore be highly context-dependent and range from antagonistic to co-operative in different cell types and genomic regions. The moderately positive correlation between BRG1 and EZH2 in stable compartments is therefore not surprising.

Minor points:

1. Page 4, statement: "BRG1, EZH2, and H3K27me3 were much more highly correlated with each other in KO cells as compared to WT cells (Figure S4B)". There is no explanation about why BRG1, EZH2, and H3K27me3 should correlate with each other. What's the purpose of this analysis?

This analysis provides a broad overview of the relationships between different epigenetic marks in WT and KO cells and shows how these relationships change upon β -actin loss. The increase in correlation signifies the dysregulation of the BAF-PRC relationship in KO cells which is explored in more detail in Figure 5A. We have slightly modified the text to clarify this point.

2. Page 5, statement: "Similar to our observations for sites showing increased accessibility in KO cells, GO term analysis of regions gaining EZH2 peaks revealed an enrichment of biological processes linked to skeletal system and vasculature development (Figure 2F and Figure 1E)". Besides GO term, please also provide the list of genes showing overlap between EZH2 peaks and ATAC-seq peaks.

We have included a list of peaks overlapping between ATAC-Seq and EZH2 CHIP-Seq and associated gene annotations in Supplementary Table S10.

3. Page 5, there's no figure S3B and S3D. Please clarify the labelling.

We thank the reviewer for pointing this out and have modified the text. The correct figure is S2B.

4. As Hi-C resolution highly depends on the quality of the data and the author try to use different resolution to analysis the data, please provide a summary of the Hi-C data, such as raw read pair number, valid pair number, Intra-chromosome read pair number and Inter-chromosome read pair number.

We have included the relevant statistics in Supplementary Table S11

5. Page 6, statement: "As compartment identity is reflected in the sign of the first principal component (PC1)," Please provide the reference that support this statement.

We have added the relevant reference.

6. The figure order is not well organized. I had to jump from different figures when I read the paper.

We appreciate that some parts of the discussion refer to data from multiple figures. However, due to the large amount of data we were unable to include all information relevant to a topic in individual figures. We have, nevertheless, tried our best to improve the figure order in the revised manuscript.

7. Page 7, statement: "TAD analysis at 50kb resolution revealed that a majority of TAD boundaries showed little or no difference in insulation scores between WT and KO cells (Figure 3G and Figure 4A)." Fig 4A is totally unrelated to insulation scores.

We thank the reviewer for pointing this out and have removed the reference to Figure 4A

8. Page 10, statement: "Since H3K9me3 is characteristic of HP1 α positive heterochromatin and shows a strong correlation with compartment switching in β -actin KO cells.". Since the R2 is only 0.37, I am not sure why the authors believe it is a strong correlation.

The statement was meant to highlight the fact that H3K9me3 shows a stronger correlation with compartment switching in comparison to other marks like H3k27me3, BRG1 and EZH2. We have modified the sentence to better reflect this.

9. Methods, no input information is provided for ATAC-Seq and ChIP-seq.

We thank the reviewer for bringing this to our attention and have included this information in the methods section. Briefly, ChIP-seq input samples were processed using the same methods as immunoprecipitated samples and used as controls for peaks calling. Input sample data is included in the relevant GEO repositories. No input samples were included for ATAC-Seq experiments as they do not require an input control.

10. Figure S10 is not mentioned in the main text.

We thank the reviewer for pointing this out and have removed Figure S10 from the supplemental figures as it does not directly relate to the results and only shows quality control data.

11. The number of replicates for each experiment needs to be mentioned.

We have now included the number of replicates in the relevant Methods subsection (Hi-C and ATAC-Seq experiments were performed using two biological replicates per condition).

12. Gene list for each GO term needs to be provided.

We included gene lists for all figures containing go-term analyses in supplementary table S12

Reviewer #3:

The relatively recent discovery of actin in the nucleus opens a very exciting field of research. Furthermore the hierarchical folding of genome folding is central to gene expression, cellular development and function. The submitted manuscript by Syed Raza Mahmoud and colleagues is in these lines of research, to understand the function of β -actin in chromatin remodeling and the 3D structure of the genome. Using a battery of genome-wide techniques including ATAC-seq, Chip-seq, RNA-seq and HiC, in mouse embryonic fibroblast (MEF) cells completely or partially deleted for β -actin, the authors find that changes in actin levels are associated with different chromatin accessibility and changes in the 3D structure of the genome. The authors propose a new role for β -actin in reshaping the genomic landscape that could be important in the regulation of genes involved in differentiation and development.

The exploitation of genome-wide data and their smart analyses are clearly assets for a global understanding of the modifications induced by changes in actin levels. However, in my opinion, the conclusions should be supported as much as possible by orthogonal approaches. This would give a definitely more solid character to the final message.

We thank the reviewer for the constructive suggestions. We have tried to reinforce our conclusions with additional experiments and bioinformatic analyses in the revised manuscript.

To begin with, the authors work on β -actin $-/-$ or β -actin $-/+$ heterozygous MEFs whose actin level of expression is verified by RNA seq. It would be good for the authors to

- i) specify how the KO was constructed and verified,

The KO cells were obtained from the lab of Dr. Christoph Ampe, University of Gent. The construction and verification of the KO is described in (Tondeleir et al., 2012).

- ii) confirm RNA-seq result by RT-QPCR and provide a WB in order to confirm the intermediate level of β -actin in heterozygous cells (not shown in Tondeleir et al., 2012).

We have previously demonstrated (Xie et al 2018) changes in the expression of different actin isoforms using RT-qPCR in WT, KO and HET cells and have also shown β -actin protein levels by immunoblotting. We are reproducing below the relevant figures from that study for the reviewer's reference.

[REDACTED]

[REDACTED]

Western blot analysis of β -actin in WT, KO and HET cells normalized to GAPDH

Figure-8, Xie et al. ,2018

qPCR analysis of β -actin in WT, KO and HET cells normalized to Nono

Figure-8, Xie et al. ,2018

The group has already shown in a previous study that the KO of β -actin has an effect on the nuclear position of the epigenetic heterochromatin constitutive mark H3K9me3 (by IF), that the level of this mark varies (by ChIP-seq) and that these effects are accompanied by an increase or decrease in various genes expression (Xie et al., 2018). In the present study the authors further correlate these changes in expression with differences in chromatin accessibility (ATAC-seq). It would be useful to clarify whether the up and down regulated genes are the same in both studies and also to confirm the absence of cleavage bias by Tn5 by testing, for example, the positioning of nucleosomes for a few positions in the WT and KO strain.

The present study uses the RNA-Seq counts from the dataset previously published in (Xie et al., 2018). While the overall differential expression results would therefore be similar, there would be some differences due to the use of slightly different cutoffs.

We have included nucleosome plots showing all TSSs in the genome and sets of randomly selected regions in supplementary figure S3B. Our result show largely similar nucleosome occupancy between WT and KO cells.

It is interesting that among the regions that have lost accessibility some correspond to TSS generally enriched with chromatin remodelers such as BAF/BRG1. The authors have already shown by ChIP-seq that the KO of β -actin leads to a loss of BRG1 on chromatin (Xie et al., 2018). This supposes a direct interaction between BRG1 and β -actin that would benefit from being shown by co-IP in the present study.

We have now performed a Co-IP experiment between β -actin and BRG1 in nuclear and cytoplasmic fractions of WT and KO cells (Figure 1A). Our results confirm a direct interaction between BRG1 and β -actin in the nucleus.

Why is ChIP-seq in heterozygous cells not shown for BRG1? (Figure 2A).

The ChIP-Seq data for BRG1, H3K27m3 and H3K9me3 were obtained from our previous study (Xie et al., 2018) which focused on WT and KO cells and no ChIP-Seq experiments was therefore performed in HET cells for these marks. While new ChIP-Seq experiments in the HET cells are ongoing in our lab, we believe that data from these experiments would not be comparable to existing WT and KO data due to batch effects. In order to make the plots more consistent, we have therefore omitted the heterozygous sample from the EZH2 panel in Figure 2 as well. All plots now only show the ChIP-Seq signal in WT and KO cells as the main focus of the present study is to demonstrate the relationship between β -actin levels and chromatin structure as shown by ATAC-Seq and HiC-Seq rather than dosage dependence between actin levels and specific epigenetic marks.

More generally, can the authors comment on the results obtained by Barisic et al (2019) and their own (i.e. deletion of BRG1 versus β -actin KO)?

Our results are largely consistent with the observations of Barisic et al. and show that the loss of β -actin and loss of BRG1 have similar effects on the genome. This suggests that changes in β -actin levels can be critical for BRG1 mediated chromatin remodeling. Barisic et al. show that loss of BRG1 has no impact on TAD insulation or CTCF binding but only affects compartment structure (also reported by Barutco et al 2016). In line with this observation, our data confirms that β -actin loss can trigger compartment switching but has no impact on insulation scores at TAD boundaries or at CTCF sites. Similarly, Barisic et al. show that loss of BRG1 leads to loss of REST chromatin binding. We have also performed ChIP-Seq of REST in β -actin WT and KO cells (data not included in manuscript), which confirms loss of REST binding upon β -actin loss. Importantly, our data also shows that cells heterozygous for β -actin, exhibit less pronounced genomic changes compared to β -actin KO cells which more closely resemble BRG1 KO cells. In addition to building upon the conclusions of Barisic et al., our work therefore also sheds light on the genome-wide effects of partial BRG1 loss using β -actin heterozygous cells as a model.

The authors went further in the regulatory role of actin in the chromatin configuration by making HiC not only in KO and HET, but also in add-back experiments. The authors observed global changes in the interactions between genomic regions without changing the nature of compartments A (euchromatin) and B (heterochromatin), except for 6% of the genome, which switch between the two compartments.

It would definitely be necessary to confirm by DNA FISH the switch between compartments A and B by measuring the distances between a few pairs of genes in a given compartment and comparing them between WT and KO cells.

We agree with the reviewer that DNA FISH is an excellent approach for further verifying compartment switching. However, we believe that based on the combined analysis of Hi-C Seq, RNA-Seq, ATAC-Seq and RT-PCR, our data already provides convincing evidence that compartment switching regions show the expected changes in transcription, accessibility, and epigenetic marks.

As expected the levels of expression by RNAseq confirms the transition from compartment A to compartment B (and vice-versa). It would be useful for the authors to confirm this by showing an RT-QPCR of selected genes.

We have now included the RT-qPCR analysis of selected switching genes in supplementary figure S6C

Given the HiC results, it could be interesting to calculate the probability of contact as a function of genomic distance $P(s)$ in order to define the potential effects of actin loss on the chromatin fibre structure. I also advise to make a differential analysis of the contacts between WT and KO, to quantify them more precisely (for instance see the star method from Kundu et al., 2017, Mol Cell).

We thank the reviewer for this suggestion. We have now included the probability of contact as a function of genomic distance in supplementary figure S11. While the expected contact frequency is largely similar between WT and KO cells, we observe a slight reduction in expected long range contacts upon β -actin loss.

As suggested, we have also performed a differential analysis of all contacts in WT and KO cells. The detailed results of this analysis are included as extended data (available for download at <https://drive.google.com/file/d/1EJVwLZJ1OSp30khdAw5mR6H8nIAsWAnH/view?usp=sharing>) while contact maps of selected genes showing differential interactions are included in supplementary figure S7.

Because the effect of actin on chromatin is different according to the cell cycle, it would be interesting in the future to refine this analysis on synchronized cells.

We thank the reviewer for this suggestion. Understanding the role of nuclear actin in cells undergoing differentiation or reprogramming and in cells at different stages of the cell cycle is an area of ongoing research in our lab.

REFERENCES

1. Alexander, J.M., Hota, S.K., He, D., Thomas, S., Ho, L., Pennacchio, L.A., Bruneau, B.G., 2015. Brg1 modulates enhancer activation in mesoderm lineage commitment. *Development* 142, 1418–30. <https://doi.org/10.1242/dev.109496>
2. Al-Sayegh, M.A., Mahmood, S.R., Khair, S.B.A., Xie, X., El Gindi, M., Kim, T., Almansoori, A., Percipalle, P., 2020. β -actin contributes to open chromatin for activation of the adipogenic pioneer factor CEBPA during transcriptional reprogramming. *Mol Biol Cell* 31, 2511–2521. <https://doi.org/10.1091/mbc.E19-11-0628>
3. Barisic, D., Stadler, M.B., Iurlaro, M., Schübeler, D., 2019. Mammalian ISWI and SWI/SNF selectively mediate binding of distinct transcription factors. *Nature* 569, 136–140. <https://doi.org/10.1038/s41586-019-1115-5>
4. Kadoch, C., Williams, R.T., Calarco, J.P., Miller, E.L., Weber, C.M., Braun, S.G., Pulice, J.L., Chory, E.J., Crabtree, G.R., 2017. Dynamics of BAF- Polycomb Complex Opposition on Heterochromatin in Normal and Oncogenic States. *Nat Genet* 49, 213–22. <https://doi.org/10.1038/ng.3734>
5. Lavarone, E., Barbieri, C.M., Pasini, D., 2019. Dissecting the role of H3K27 acetylation and methylation in PRC2 mediated control of cellular identity. *Nat Commun* 10, 1679. <https://doi.org/10.1038/s41467-019-09624-w>
6. Mashtalir, N., Suzuki, H., Farrell, D.P., Sankar, A., Luo, J., Filipovski, M., D’Avino, A.R., St Pierre, R., Valencia, A.M., Onikubo, T., Roeder, R.G., Han, Y., He, Y., Ranish, J.A., DiMaio, F., Walz, T., Kadoch, C., 2020. A Structural Model of the Endogenous Human BAF Complex Informs Disease Mechanisms. *Cell* 183, 802-817.e24. <https://doi.org/10.1016/j.cell.2020.09.051>
7. Matsumoto, S., Banine, F., Struve, J., Xing, R., Adams, C., Liu, Y., Metzger, D., Chambon, P., Rao, M.S., Sherman, L.S., 2006. Brg1 is required for murine neural stem cell maintenance and gliogenesis. *Developmental Biology* 289, 372–383. <https://doi.org/10.1016/j.ydbio.2005.10.044>
8. Miyamoto, K., Gurdon, J.B., 2013. Transcriptional regulation and nuclear reprogramming: roles of nuclear actin and actin-binding proteins, in: *Cell Mol Life Sci.* pp. 3289–302. <https://doi.org/10.1007/s00018-012-1235-7>
9. Nishimoto, N., Watanabe, M., Watanabe, S., Sugimoto, N., Yugawa, T., Ikura, T., Koiwai, O., Kiyono, T., Fujita, M., 2012. Heterocomplex formation by Arp4 and β -actin is involved in the integrity of the Brg1 chromatin remodeling complex. *J Cell Sci* 125, 3870–3882. <https://doi.org/10.1242/jcs.104349>
10. Ooi, L., Belyaev, N.D., Miyake, K., Wood, I.C., Buckley, N.J., 2006. BRG1 Chromatin Remodeling Activity is Required for Efficient Chromatin Binding by the Transcriptional Repressor Rest and Facilitates Rest-Mediated Repression. *J Biol Chem* 281, 38974–38980. <https://doi.org/10.1074/jbc.M605370200>

11. Tondeleir, D., Lambrechts, A., Müller, M., Jonckheere, V., Doll, T., Vandamme, D., Bakkali, K., Waterschoot, D., Lemaistre, M., Debeir, O., Decaestecker, C., Hinz, B., Staes, A., Timmerman, E., Colaert, N., Gevaert, K., Vandekerckhove, J., Ampe, C., 2012. Cells Lacking β -Actin are Genetically Reprogrammed and Maintain Conditional Migratory Capacity*. *Mol Cell Proteomics* 11, 255–271. <https://doi.org/10.1074/mcp.M111.015099>
12. Xie, X., Almuzzaini, B., Drou, N., Kremb, S., Yousif, A., Farrants, A.O., Gunsalus, K., Percipalle, P., 2018. beta-Actin-dependent global chromatin organization and gene expression programs control cellular identity. *FASEB J* 32, 1296–1314. <https://doi.org/10.1096/fj.201700753R>
13. Young, M.D., Willson, T.A., Wakefield, M.J., Trounson, E., Hilton, D.J., Blewitt, M.E., Oshlack, A., Majewski, I.J., 2011. ChIP-seq analysis reveals distinct H3K27me3 profiles that correlate with transcriptional activity. *Nucleic Acids Res* 39, 7415–7427. <https://doi.org/10.1093/nar/gkr416>
14. Zhao, K., Wang, W., Rando, O.J., Xue, Y., Swiderek, K., Kuo, A., Crabtree, G.R., 1998. Rapid and phosphoinositol-dependent binding of the SWI/SNF-like BAF complex to chromatin after T lymphocyte receptor signaling. *Cell* 95, 625–36.

REVIEWER COMMENTS

Reviewer #1 (Remarks to the Author):

The authors have made a great effort to address the issues already brought forward. Typically both through changes in text but also with new experiments supporting their claims. I have no other concerns that are not addressed by the other reviewers.

Sandra Peiró, PhD
Head of Chromatin Dynamics in Cancer Group
Vall d'Hebron Institute of Oncology (V.H.I.O.)
Edificio Cellex
c/ Natzaret 115-117,
08035 Barcelona Spain

Reviewer #2 (Remarks to the Author):

Overall, the authors have responded well to many of my major concerns. The author also provides additional CO-IP evidence to support β -actin can interact with BRG1 in the nucleus. There are, however, a couple of issues remaining to be addressed as outlined below.

1. The authors gave a reasonable explanation for why NLS-mouse/human rescue is subtle. The expression of β -actin in the differentiated cells can't revert the chromatin change occurring during differentiation. The comparisons among biological replicates also show consistency of the rescue experiments. It would be good to show that some of the un-rescued areas is due to differentiation.
2. The authors didn't repeat the WB results from their previous study. The WB showed in the response is just a copy from the original manuscript. Their results do show the intermediate β -actin level in Heterozygous. However, studies from other groups (reference 8: Tondeleir D et. al., Mol Cell Proteomics. 2012, Fig 1F) do not show such reduction of the protein in heterozygous (Figure copied as below). As the protein level in the heterozygous is important to draw the dosage-dependent conclusion, it is critical to repeat such an easy experiment.

Reviewer #3 (Remarks to the Author):

I thank the authors for their effort in responding to many of my suggestions and comments. For instance, the Co-IP shown in Figure 1A definitely gives more weight to this study.

However, I am skeptical about why the authors do not show new RT-QPCR on the cells used in this study.

I believe ATAC-seq and HiC are new data made on the previously published strain: the verification of the cells used for these new set of experiments is mandatory and we cannot be satisfied with the blots from Xie et al 2018. The 3 reviewers have made this same request and it seems to me that the answer is not up to expectations. In this experiment, the add-back strain, much discussed by Referee 1 would also be a plus.

As for my request for DNA FISH, it seems important to me to validate the results of HiC and I do not understand why it is not done, including by a possible collaborator. This would be an excellent way to validate the B>A and A>B switch claimed by the authors, given that the changes in expression validated on certain genes as I had requested remain modest (p-values based on two-tailed Welch t-test, *p<0.05)

Rebuttal letter – manuscript no. NCOMMS-20-25611-B

We would like to thank all reviewers for their constructive feedback. Please find below our point-by-point responses (black text) to each of the reviewers' concerns (red text).

Reviewer #1 (Remarks to the Author):

The authors have made a great effort to address the issues already brought forward. Typically both through changes in text but also with new experiments supporting their claims. I have no other concerns that are not addressed by the other reviewers.

We thank the reviewer for the constructive feedback which helped us significantly improve the manuscript.

Reviewer #2 (Remarks to the Author):

Overall, the authors have responded well to many of my major concerns. The author also provides additional CO-IP evidence to support β -actin can interact with BRG1 in the nucleus. There are, however, a couple of issues remaining to be addressed as outlined below.

We thank the reviewer for the comments and suggestions and have tried to address the remaining concerns below

1. The authors gave a reasonable explanation for why NLS-mouse/human rescue is subtle. The expression of β -actin in the differentiated cells can't revert the chromatin change occurring during differentiation. The comparisons among biological replicates also show consistency of the rescue experiments. It would be good to show that some of the un-rescued areas is due to differentiation.

We have now included a list of genes overlapping un-rescued switching bins in supplementary table-S14. While the un-rescued genomic bins contain many genes involved in differentiation, given the large number of diverse genes present in these regions we cannot assert that all the un-rescued bins are involved in differentiation. A GO-term enrichment analysis, however, does include many developmental processes and at least partly supports the idea that some of these bins may regulate development and differentiation.

2. The authors didn't repeat the WB results from their previous study. The WB showed in the response is just a copy from the original manuscript. Their results do show the intermediate β -actin level in Heterozygous. However, studies from other groups (reference 8: Tondeleir D et. al., Mol Cell Proteomics. 2012, Fig 1F) do not show such reduction of the protein in heterozygous (Figure copied as below). As the protein level in the heterozygous is important to draw the dosage-dependent conclusion, it is critical to repeat such an easy experiment.

We have now repeated the western blot and RT-QPCR of β -actin in WT, KO and HET cells and included it in Figure 1B. Our results are consistent with the western blot shown in Xie et al. 2018 and confirm a reduction in β -actin levels in HET cells compared to WT cells.

Reviewer #3 (Remarks to the Author):

I thank the authors for their effort in responding to many of my suggestions and comments. For instance, the Co-IP shown in Figure 1A definitely gives more weight to this study.

We appreciate the reviewer's comments and have tried to address some of them below.

However, I am skeptical about why the authors do not show new RT-QPCR on the cells used in this study. I believe ATAC-seq and HiC are new data made on the previously published strain: the verification of the cells used for these new set of experiments is mandatory and we cannot be satisfied with the blots from Xie et al 2018. The 3 reviewers have made this same request and it seems to me that the answer is not up to expectations. In this experiment, the add-back strain, much discussed by Referee 1 would also be a plus.

We had previously reproduced western blots and RT-QPCR of β -actin levels from Xie et al 2018 as the present work not only utilizes the same cells but also analyzes ChIP-Seq and RNA-Seq data from that study. However, to address the reviewers concern we have now repeated the western blot and RT-QPCR experiments and included them in Figure 1B. Our results confirm a reduction in β -actin levels in HET cells compared to WT cells. As per the reviewer's suggestion we are also including below a western blot of the rescue sample expressing NLS- β -actin. It should be noted that due to the presence of the HA tag at the binding site of the β -actin antibody, β -actin expression in the rescue sample has been shown using an HA-tag antibody. Furthermore, we have included additional western blots in Supplementary Figure S4C showing the direct effect of β -actin levels on histone methylation.

As for my request for DNA FISH, it seems important to me to validate the results of HiC and I do not understand why it is not done, including by a possible collaborator. This would be an excellent way to validate the B>A and A>B switch claimed by the authors, given that the changes in expression validated on certain genes as I had requested remain modest (p-values based on two-tailed Welch t-test, *p<0.05)

We appreciate the reviewer's point regarding reconciling HiC results with DNA-FISH and have dedicated considerable effort in setting up DNA-FISH experiments over the last few months. While we were able to target selected switching and non-switching loci using fluorochrome labelled BAC probes, our imaging results were not precise enough to reliably quantify changes in spatial distances between switching regions. However, we believe that our conclusions regarding actin-dependent compartment switching

are independent of this result as they are based on experimentally validated changes in interaction frequencies measured using widely accepted HiC protocols. Although Hi-C data is generally known to be in agreement with interactions inferred from DNA FISH, deviations from this trend can arise due to various technical and detection biases inherent to both assays and a fraction of variance in either technique remains unexplained by the other (R.P. McCord, N. Kaplan, L. Giorgetti, 2020). Furthermore, concordance between the two techniques requires genomic loci to be within a few hundred nanometers and is highest when spatial distance thresholds of 120–150 nm are met (R.P. McCord, N. Kaplan, L. Giorgetti, 2020). As a result several studies have urged caution in directly correlating HiC interaction frequencies with DNA-FISH without validation from other independent techniques (Williamson, Iain, et al., 2014, Fudenberg, Geoffrey, and Maxim Imakaev, 2017). We therefore believe that evidence from independent sequencing experiments such as RNA-Seq and ATAC-Seq gives added weight to our HiC analysis by demonstrating the functional impact of compartment level changes on the transcriptional and chromatin accessibility landscape. We are including below a differential analysis showing volcano plots of genes and ATAC-seq peaks overlapping switching regions. This analysis clearly shows that regions predicted to switch from B to A or A to B compartments by HiC correlate with gene activation and repression respectively while non-switching regions show no such relationship with transcription or accessibility. While we appreciate the importance of correlating HiC interactions with DNA-FISH, we believe that an in-depth analysis of the reasons due to which HiC and imaging analysis may not agree at specific loci would entail an independent investigation in itself. We hope that the additional RNA-Seq and ATAC-Seq based validation of HiC data will at least partly address the reviewer’s concerns.

REVIEWERS' COMMENTS

Reviewer #2 (Remarks to the Author):

I appreciate all the efforts that the authors have made in response to my concerns. All my concerns have already been fully addressed.

Reviewer #3 (Remarks to the Author):

I thank the authors for their efforts and response;

The new data in Figure 1B are important and ultimately reinforce the underlying basis of the entire analysis.

Regarding my second point about FISH DNA as an alternative corroborative method, the authors' arguments are valid. However, it would be welcome to argue this point in a little more detail in the discussion

REVIEWERS' COMMENTS

Reviewer #2 (Remarks to the Author):

I appreciate all the efforts that the authors have made in response to my concerns. All my concerns have already been fully addressed.

We thank the reviewer for the constructive feedback which helped us greatly improve the manuscript.

Reviewer #3 (Remarks to the Author):

I thank the authors for their efforts and response;

The new data in Figure 1B are important and ultimately reinforce the underlying basis of the entire analysis.

Regarding my second point about FISH DNA as an alternative corroborative method, the authors' arguments are valid. However, it would be welcome to argue this point in a little more detail in the discussion.

We thank the reviewer for the valuable recommendations. We have included a sentence in the discussion describing DNA-FISH analysis of switching regions as a future research direction.